# Redox regulation of G protein oligomerization and signaling by the glutaredoxin WG1 controls grain size in rice

Lijie Liu[1,2,3], Jianqin Hao[1,3], Ke Huang [ID][1], Penggen Duan[1], Baolan Zhang[1], Zhihai Chi [ID][1], Xiaohong Yao[1] & Yunhai Li [ID][1,2✉]

## Abstract

**Grain size is an important agronomic trait and influences both grain yield and quality in crops. The atypical heterotrimeric Gγ protein subunit GS3 is a central regulator of grain length in rice, and the loss-of-function allele of its corresponding gene has been widely utilized by breeders to improve grain length in rice. Here we report that the CC-type glutaredoxin WG1/OsGRX8 has disulfide oxidoreductase activity and regulates redox state of GS3, thereby determining grain length in rice. GS3 can form dimers and oligomers by intermolecular disulfide bonds, and the cysteine-rich C-terminal region of GS3 is predominantly required for its oligomerization. The oligomerization of GS3 alleviates its inhibitory effect on the interaction between RGB1 and DEP1/GGC2, resulting in an increase in grain length. WG1 interacts with GS3 and reduces the oligomerization of GS3 through redox mechanisms, which causes a decrease in grain length. Genetic analyses support WG1 and GS3 function in a common pathway to control grain length. Thus, our findings reveal a previously unrecognized mechanism, in which redox regulation of a Gγ subunit by a glutaredoxin controls grain length, opening a novel perspective for G protein signaling regulation.**

**Keywords** WG1; GS3; Grain Length; Redox Regulation; G Protein
**Subject Category** Plant Biology

## Introduction

Grain size is an important yield trait in crops. Increasing crop yields to ensure food security and sustainable agriculture is a great challenge (Tilman et al, 2011). Exploring the genetic and molecular mechanisms of grain size control will offer effective strategies for high-yield and quality breeding in crops. Several pathways that control grain size have been described in rice (Li et al, 2019). In particular, several studies have revealed the heterotrimeric guanine nucleotide-binding protein (G protein) signaling plays a crucial role in grain size regulation (Duan and Li 2021; Liu et al, 2018; Sun et al, 2018). The G protein complex consists of three subunits, Gα, Gβ, and Gγ, and presents conserved signaling mechanisms in eukaryotes. When Gα binds GDP, Gα and heterodimer Gβγ form a heterotrimer that is in an inactive state, while when GDP is exchanged for GTP, Gα separates from Gβγ and initiates their downstream effectors (Xu et al, 2016). The rice genome contains one Gα subunit (RGA1), one Gβ subunit (RGB1), and five Gγ subunits (RGG1, RGG2, GS3, DEP1, and GGC2) (Sun et al, 2018). RGA1 and RGB1 are positive regulators of grain length in rice (Sun et al, 2018). The two typical Gγ subunits (RGG1 and RGG2) negatively regulate rice grain length (Miao et al, 2019; Tao et al, 2020). By contrast, three atypical Gγ subunits (GS3, DEP1, and GGC2) have different effects on grain length possibly due to the variation of their respective cysteine (Cys)-rich C-terminus (Sun et al, 2018). DEP1 and GGC2 positively influence grain length, while GS3 negatively regulates grain length (Sun et al, 2018). It has been proposed that RGB1 interacts with DEP1 and GGC2 to promote grain growth, while GS3 inhibits the interaction of RGB1 with DEP1 and GGC2 to restrict grain growth (Sun et al, 2018). E3 ubiquitin ligase CLG1 has been described to target GS3 for degradation (Yang et al, 2021b). Importantly, the loss-of-function allele of *GS3* has been widely utilized in elite varieties to improve grain length (Zeng et al, 2019; Zhou et al, 2019). However, it is still unclear how GS3 activity is regulated to influence grain length.

Redox modifications play important roles in regulating plant growth and development (Sevilla et al, 2023; Zhou et al, 2022). The redox state is crucial for the function of proteins. For example, the thiol groups (-SH) present in Cys residues of proteins are subject to various redox post-translational modifications, which are crucial for protein structures and activities. Glutaredoxins (GRXs) are glutathione (GSH) dependent oxidoreductases that reversibly reduce disulfide bonds (-SSR) or S-glutathione (-SSG) of target proteins to -SH, thereby influencing redox homeostasis and signaling (Chai and Mieyal, 2023). The CC-type GRX protein WG1/OsGRX8 is a regulator of grain size, and can be ubiquitinated and targeted for degradation by the E3 ubiquitin ligase GW2 (Hao et al, 2021). Meanwhile, WG1 can inhibit the transcriptional activity of OsbZIP47 by recruiting the transcriptional co-repressor ASP1 to regulate grain size and weight (Hao et al, 2021). WG1 has been demonstrated to have disulfide oxidoreductase activity (Hao

[1]State Key Laboratory of Seed Innovation, Institute of Genetics and Developmental Biology, Chinese Academy of Sciences, Beijing, China. [2]College of Advanced Agriculture, University of Chinese Academy of Sciences, Beijing, China. [3]These authors contributed equally: Lijie Liu, Jianqin Hao. ✉E-mail: yhli@genetics.ac.cn

et al, 2021). However, the specific substrates targeted by WG1 in grain size control remain unknown.

To understand how WG1 exerts oxidoreductase activity to control grain size, we identify WG1-interacting proteins. We find that WG1 physically interacts with GS3. GS3 contains active –SH that facilitates the formation of dimers and oligomers through intermolecular disulfide bonds. The oligomeric state of GS3 alleviates its inhibitory effect on the interaction between RGB1 and DEP1/GGC2, thereby promoting grain growth. WG1 has the capacity to reduce the dimers and oligomers of GS3, decreasing grain length. Genetic analyses support that WG1 and GS3 have overlapped function in grain length control. Thus, our results define a previously unknown mechanism that the redox regulation of GS3 by WG1 controls grain length, opening a perspective for the G protein signaling regulation.

# Results

## WG1/OsGRX8 physically interacts with GS3

We previously revealed that the CC-type glutaredoxin WG1 promotes grain growth in the grain-width direction, but represses grain growth in the grain-length direction (Hao et al, 2021). Considering that WG1 has disulfide oxidoreductase activity (Hao et al, 2021), we tried to understand how WG1 exerts oxidoreductase activity to control grain size. We previously identified WG1-interacting proteins using a yeast two-hybrid (Y2H) screen (Hao et al, 2021). One of WG1-interacting proteins is GS3, an atypical γ-subunit of the G-protein. Because *GS3* is a major QTL gene for grain length, and its loss-of-function allele was widely used in elite varieties to improve grain length (Zeng et al, 2019; Zhou et al, 2019), we tried to understand the molecular and genetic relationships between WG1 and GS3 in grain size control.

We first tested whether WG1 could interact with the full-length GS3 in yeast cells. The full-length coding sequence (CDS) of *GS3* from ZH11 was fused with GAL4 activation domain (AD) to generate AD-GS3. Yeast cells co-transformed with GAL4 DNA binding domain (BD)-WG1 and AD-GS3 grew well on the selective media, while no yeast cells grew on the selective media for the negative controls (Fig. 1A), indicating that WG1 interacts with the full-length GS3 in yeast cells. Subsequently, we conducted a pull-down assay to investigate whether WG1 could directly interact with GS3. MBP-GS3 and FLAG-WG1 fusion proteins were expressed in *Escherichia coli* (*E. coli*) cells and then incubated with MBP beads to immunoprecipitate the MBP-GS3. As shown in Fig. 1B, FLAG-WG1 was detected in the immunoprecipitated MBP-GS3, but not in the negative control (MBP), indicating that WG1 physically interacts with GS3 in vitro.

We then tested whether GS3 could interact with WG1 in vivo using firefly luciferase complementation imaging (LCI) assays. We fused GS3 to the C-terminus of luciferase to generate cLUC-GS3. cLUC-GS3 was co-transformed with WG1-nLUC in *Nicotiana benthamiana* (*N. benthamiana*) leaves, while co-transformation of cLUC-GS3 with EOG1-nLUC and WG1-nLUC with cLUC-EOG1 served as negative controls (Yan et al, 2024). Luciferase activity was observed when we co-expressed cLUC-GS3 with WG1-nLUC, while the negative controls showed no detectable luciferase activity (Fig. 1C), indicating that WG1 associates with GS3 in vivo. We

further adopted the bimolecular fluorescence complementation (BiFC) assays to investigate the interaction between GS3 and WG1. We transiently co-expressed cYFP-GS3 or cYFP (C-terminal fragment of YFP) with nYFP-WG1 or nYFP (N-terminal fragment of YFP) along with PIP2-mCherry, which served as a plasma membrane marker, in *N. benthamiana* leaves, respectively. Strong yellow fluorescent protein (YFP) fluorescence was observed around the plasma membrane region and the cytoplasm when cYFP-GS3 was co-expressed with nYFP-WG1, whereas no YFP fluorescence was detected in the negative controls (Fig. 1D). A previous result showed that GS3 is localized in the plasma membrane, cytoplasm and nucleus (Liu et al, 2018). We then investigated whether WG1 is able to localize in the cytoplasm. We separated the cytoplasmic and nuclear protein fractions from *proWG1:WG1-GFP* transgenic plants. WG1-GFP proteins were detected in both the cytoplasmic and the nuclear protein fractions (Appendix Fig. S1). These results showed that WG1 is localized in both the cytoplasm and the nucleus, although the WG1-GFP signal was strongly observed in the nuclei, but not strongly detected in the cytoplasm (Hao et al, 2021). Thus, these results supported that WG1 associates with GS3 in the cytoplasm adjacent to the plasma membrane as well as other parts of the cytoplasm.

We further asked whether GS3 could associate with WG1 in rice plants. The *pro35S:MYC-WG1* and *pro35S:GFP-GS3* rice transgenic lines were generated in ZH11 plants. We crossed *pro35S:MYC-WG1* with *pro35S:GFP* or *pro35S:GFP-GS3* transgenic lines to obtain *pro35S:GFP;pro35S:MYC-WG1* and *pro35S:GFP-GS3;pro35S:MYC-WG1* plants, respectively. Total proteins were extracted from young panicles of these plants and incubated with GFP-Trap® Agarose beads for immunoprecipitation of GFP or GFP-GS3 complex, respectively. As shown in Fig. 1E, MYC-WG1 was detected in the immunoprecipitated GFP-GS3 complex in panicles, but was absent in the negative control (GFP) (Fig. 1E), indicating that GS3 physically associates with WG1 in rice plants.

## GS3 can oligomerize via intermolecular disulfide bonds

Given that glutaredoxin WG1 has disulfide oxidoreductase activity and interacts with GS3 that contains Cys-rich sequence in its C-terminus (Fig. EV1A), we asked whether GS3 could form intermolecular disulfide bonds through these Cys residues. The presence of active thiol groups is a prerequisite for intermolecular disulfide bond formation. We therefore performed a biotin conjugated iodoacetamide (BIAM)-labeling assay to test whether GS3 contains active thiol groups. BIAM and $H_2O_2$ can competitively react with active thiol groups. The active thiol groups in proteins can be determined through semi-quantitative detection of BIAM labeled target protein using immunoblot (Fig. 2A). We expressed MBP-GS3 in *E. coli* cells. Equal amounts of purified MBP-GS3 were incubated with different concentrations of $H_2O_2$, and followed by the reaction with BIAM. The total protein and the BIAM-labeled protein were detected using anti-MBP and anti-Biotin antibodies, respectively. We observed that MBP-GS3 was susceptible to labeling with BIAM, and this labeling was found to diminish with a gradual increase in $H_2O_2$ concentrations (Fig. 2B). In contrast, the negative control MBP-FLAG did not exhibit any labeling under similar conditions (Fig. 2B). These results supported that GS3 possesses active thiol groups.

The presence of active thiol groups in GS3 suggested that GS3 may form intermolecular disulfide bonds that can make GS3 self-

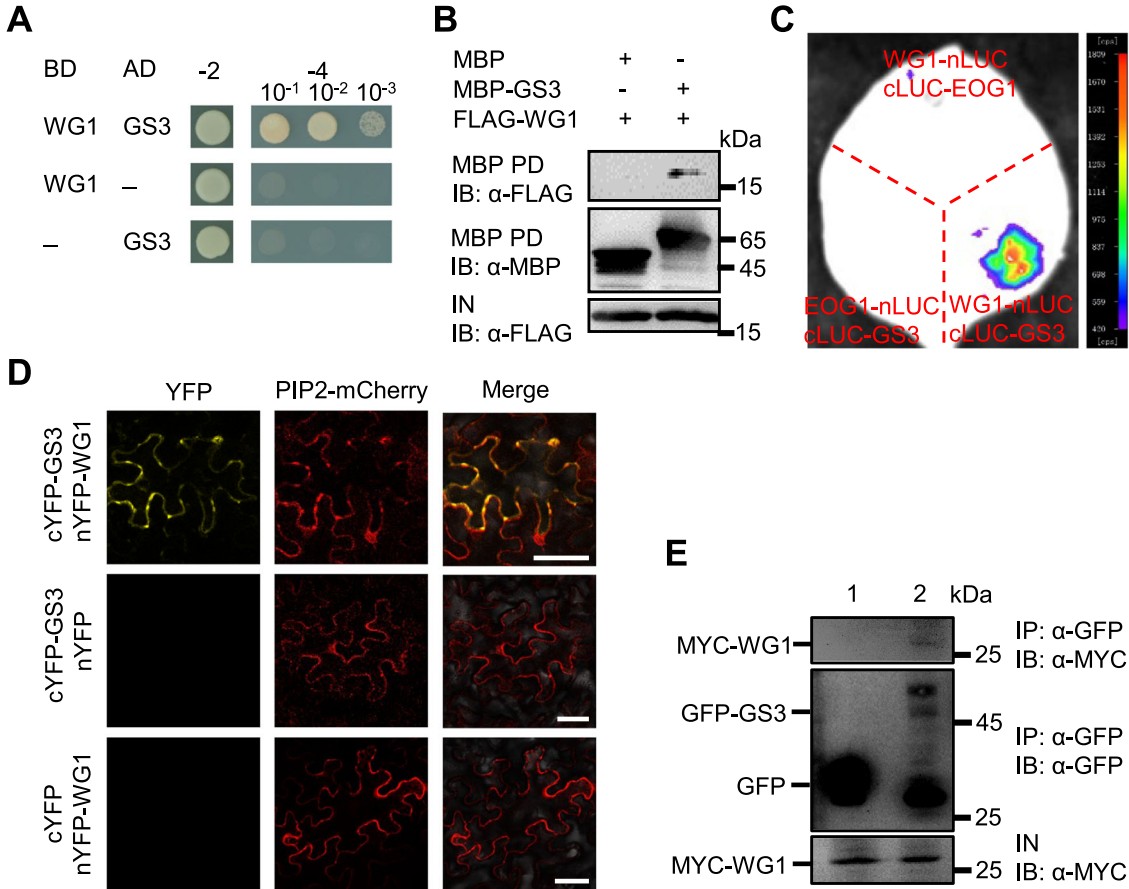

**Figure 1. WG1 interacts with GS3.**

(A) WG1 interacts with GS3 in yeast cells. The indicated construct pairs were co-transformed into yeast strain AH109. Interactions between bait and prey were examined on the control media SD-2 (SD/-Leu/-Trp) and selective media SD-4 (SD/-Leu/-Trp/-His/-Ade). BD-WG1/AD and AD-GS3/BD pairs are negative controls. (B) WG1 interacts with GS3 in pull-down assay. MBP-GS3 and FLAG-WG1 fusion proteins were expressed and incubated with MBP beads. Precipitates were detected with anti-MBP and anti-FLAG antibodies, respectively. MBP is a negative control. PD, pull-down; IB, immunoblot; IN, input. (C) WG1 associates with GS3 in firefly luciferase complementation imaging (LCI) assay. *N. benthamiana* leaves were transformed by injection of *Agrobacterium* GV3101 cells harboring *WG1-nLUC* and *cLUC-GS3* plasmids. Co-transformation of *WG1-nLUC* with *cLUC-EOG1* and *EOG1-nLUC* with *cLUC-GS3* served as negative controls. Strong luciferase complementation signal was observed for *WG1-nLUC* and *cLUC-GS3* combination, while no obvious signal was observed for the negative controls. (D) WG1 associates with GS3 in BiFC assays in *N.benthamiana* leaves. cYFP-GS3/nYFP-WG1, nYFP-WG1/cYFP and cYFP-GS3/nYFP were co-expressed with plasma membrane marker (PIP2-mCherry) in leaves of *N.benthamiana*, respectively. Strong YFP fluorescence was observed around the plasma membrane region and the cytoplasm for cYFP-GS3/nYFP-WG1 combination, while no obvious signal was observed for the negative controls. Scale bars represent 50 μm. (E) WG1 associates with GS3 in rice. *pro35S:GFP-GS3;pro35S:MYC-WG1* and *pro35S:GFP;pro35S:MYC-WG1* transgenic rice panicles were used to perform co-immunoprecipitation assay. Total proteins from *pro35S:GFP; pro35S:MYC-WG1* (1) and *pro35S:GFP-GS3;pro35S:MYC-WG1* (2) panicles (10–15 cm) were isolated and incubated with GFP-Trap® Agarose beads, and precipitates were detected with anti-GFP and anti-MYC antibodies, respectively. MYC-WG1 were detected in the immunoprecipitated GFP-GS3 complex. IP immunoprecipitation, IB immunoblot, IN input. Source data are available online for this figure.

interact. Subsequently, we validated GS3's self-interaction using Y2H, LCI, and BiFC assays. The full-length CDS of *GS3* from ZH11 was fused with BD to generate BD-GS3. Yeast cells co-transformed with AD-GS3 and BD-GS3 exhibited robust growth on selective media, while the negative controls did not grow on the selective media (Fig. 2C), indicating that GS3 could indeed self-interact in yeast cells. To determine which regions of GS3 are responsible for its self-interaction, we generated two truncated forms of GS3 for Y2H assays according to a previous report (Sun et al, 2018). GS3$^{1-94}$ contains the N-terminal Gγ-like (GGL)/organ size regulation (OSR) domain, while GS3$^{95-232}$ possesses the C-terminal Cys-rich domain (also called the "Cys-rich tail") (Fig. 2D). As shown in Fig. 2D, both yeast cells carrying the BD-GS3 and AD-GS3$^{1-94}$

combination and the BD-GS3 and AD-GS3$^{95-232}$ combination grew on selective media. The latter combination exhibited much superior growth compared with the former, and the relative β-galactosidase activity was significantly stronger (Fig. 2E). These results suggested that GS3 can self-interact through its N-terminal and C-terminal regions, and the C-terminal region has much stronger self-interaction capability. We further verified that GS3 could interact with itself in *N. benthamiana* leaves using LCI assay (Fig. 2F). Subsequently, we conducted BiFC analyses to investigate GS3 self-interaction. Co-expression of cYFP-GS3 with nYFP-GS3 resulted in strong YFP fluorescence in the plasma membrane and the cytoplasm of epidermal cells in *N. benthamiana* leaves. This fluorescence partially co-localized with the plasma membrane

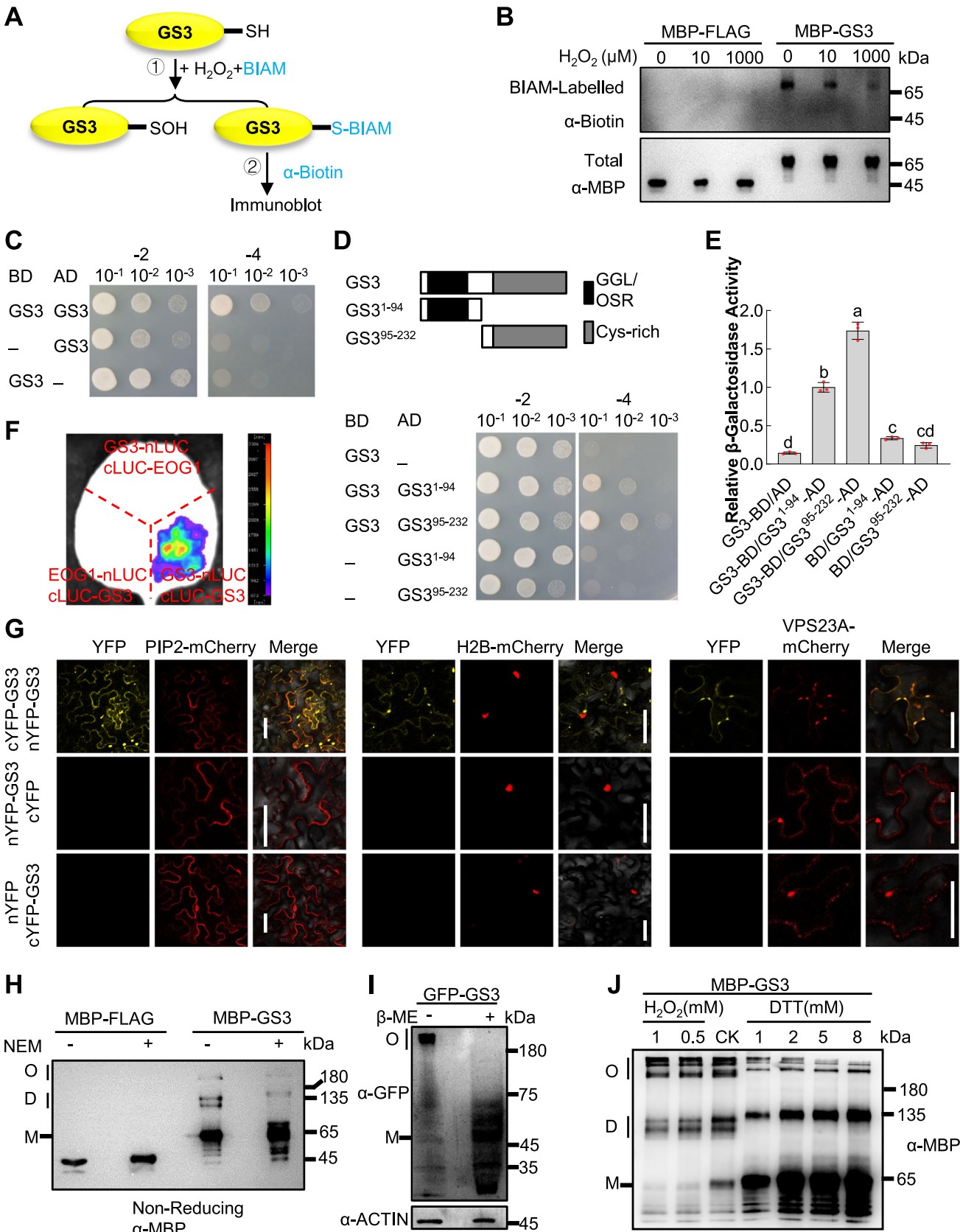

◀

**Figure 2. GS3 can oligomerize via intermolecular disulfide bonds.**

(A) Flow diagram of BIAM-labeling assay detecting the presence of active thiol groups in GS3. MBP-GS3 proteins were expressed and purified, then treated with different concentrations of $H_2O_2$ and followed by incubation with the same amount of BIAM, which could react with active thiol groups (①). S-BIAM represents the thiol labeled with BIAM. BIAM-labeled MBP-GS3 was detected by an anti-Biotin antibody (②). (B) The BIAM-labeling assay confirmed the presence of active thiol groups in GS3. According to the flow diagram outlined in (A), with an increase in the concentration of $H_2O_2$, the level of BIAM-labeled MBP-GS3 proteins progressively decreased. In contrast, the purified MBP-FLAG protein (a negative control) did not exhibit any BIAM-labeled bands. To ensure uniformity of protein levels in each reaction system, both MBP-GS3 and MBP-FLAG in the reaction mixtures were detected using an anti-MBP antibody. All the proteins were separated on 10% reducing SDS-PAGE gels. (C, D) GS3 can interact with itself in yeast cells. The protein structure of GS3 is shown. The N-terminus of GS3 (aa:1–94) including GGL/OSR domain and the C-terminus (aa:95–232) including Cys-rich domain are shown (D). The indicated construct pairs were co-transformed into yeast strain AH109. Interactions between bait and prey were examined on the control media SD-2 (SD/-Leu/-Trp) and selective media SD-4 (SD/-Leu/-Trp/-His/-Ade). (E) The relative β-galactosidase activity was quantified for each pair of bait and prey proteins as indicated in (D). The values are means ± SD ($n = 3$). The average value of GS3-BD/GS3[1-94]-AD pair was set at 1. Different lowercase letters denote significant differences among the various pairs, as determined by one-way ANOVA with Tukey's multiple comparisons test. $p = 0.0189$ (GS3-BD/AD vs. BD/GS3[1-94]-AD), 0.3062 (GS3-BD/AD vs. BD/GS3[95-232]-AD), 0.3951 (BD/GS3[1-94]-AD vs. BD/GS3[95-232]-AD), <0.0001 for other comparisons. (F) GS3 can interact with itself in LCI assay. *N.benthamiana* leaves were transformed by injection of *Agrobacterium* GV3101 cells harboring *GS3-nLUC* and *cLUC-GS3* plasmids. Co-transformation of *GS3-nLUC* with *cLUC-EOG1* and *EOG1-nLUC* with *cLUC-GS3* served as negative controls. Strong luciferase complementation signal was observed for *GS3-nLUC* and *cLUC-GS3* combination, while no obvious signal was observed for the negative controls. (G) GS3 interacts with itself in BiFC assays in *N.benthamiana* leaves. cYFP-GS3/nYFP-GS3, nYFP-GS3/cYFP and cYFP-GS3/nYFP were co-expressed with the plasma membrane marker (PIP2-mCherry), nuclear marker (H2B-mCherry) or endosome marker (VPS23A-mCherry) in leaves of *N.benthamiana*, respectively. Strong YFP fluorescence was observed in plasma membrane and endosome for cYFP-GS3/nYFP-GS3 combination, while no obvious signal was observed for the negative controls. Scale bars represent 50 μm. (H) GS3 associates with itself through intermolecular disulfide bonds in vitro. Purified MBP-GS3 and MBP-FLAG proteins, treated with or without NEM (N-ethylmaleimide), were separated on a 10% non-reducing SDS-PAGE gel, and subsequently immunoblotted using an anti-MBP antibody. Bands representing monomers, dimers and oligomers of MBP-GS3 were clearly observed in the absence of NEM. The oligomerization capability of MBP-GS3 decreased with the addition of NEM. In contrast, the negative control MBP-FLAG exclusively exhibited monomeric forms regardless of NEM treatment. (I) GS3 can form oligomers in rice plants. GFP-GS3 proteins extracted from 10–15 cm young panicles of *pro35S:GFP-GS3* transgenic rice plants were separated on a 10% non-reducing SDS-PAGE gel. The presence of GFP-GS3 oligomers in rice plants was evident, and these oligomers could be disrupted by 30% β-Mercaptoethanol (β-ME) treatment. ACTIN protein was separated on a 10% reducing SDS-PAGE gel and detected with an anti-ACTIN antibody as a loading control. (J) The conformation of GS3 is subject to redox regulation by DTT and $H_2O_2$. Purified MBP-GS3 proteins, treated with varying concentrations of $H_2O_2$ and DTT, were separated on an 8% non-reducing SDS-PAGE gel. With increasing DTT concentration, there was a gradual increase in the levels of monomers and dimers, accompanied by a decrease in the levels of oligomers. The addition of $H_2O_2$ led to a decrease in monomers and dimers, while the levels of oligomers increased proportionally with the concentration of $H_2O_2$. M, D, and O in (H–J) denote monomers, dimers, and oligomers, respectively. Source data are available online for this figure.

marker (PIP2-mCherry) and endosome marker (VPS23A-mCherry), but did not co-localize with the nuclear marker (H2B-mCherry) (Fig. 2G). These results showed that GS3 can interact with itself in the plasma membrane and cytoplasm.

To test whether GS3 could associate with itself through intermolecular disulfide bonds, we expressed MBP-GS3 in *E. coli* cells, loaded the purified MBP-GS3 with a non-reducing loading buffer onto a SDS–PAGE (sodium dodecyl sulfate-polyacrylamide gel electrophoresis) gel, and analyzed the conformation of MBP-GS3 using an immunoblot with an anti-MBP antibody. As shown in Fig. 2H, MBP-GS3 showed several discretely sized bands, supporting the existence of monomers, dimers, and oligomers of GS3. Upon adding NEM (N-ethylmaleimide, a reagent that irreversibly blocks free thiols), oligomers and dimers of GS3 were significantly decreased, while the population of monomers was notably increased. In contrast, the negative control MBP-FLAG maintained a monomeric state regardless of NEM treatment (Fig. 2H). These results indicated that GS3 has the capacity for dimerization and oligomerization via intermolecular disulfide bonds in vitro.

We then tested whether GS3 is able to form oligomers in rice plants. Total proteins extracted from *pro35S:GFP-GS3* young panicles were separated by SDS–PAGE with or without reducing agents. The conformation of GFP-GS3 was analyzed by immunoblotting with an anti-GFP antibody. GFP-GS3 proteins showed large aggregated bands without reducing agents treatment (Fig. 2I), indicating the presence of oligomers. When 30% β-ME (beta-mercaptoethanol) was added, the oligomers were dramatically decreased, while the monomers were increased (Fig. 2I), indicating that GS3 can form oligomers through intermolecular disulfide bonds in rice.

Considering that the stability of disulfide bonds is affected by the redox environment, we then examined whether the conformation of GS3 can be affected by the redox regulation. Purified MBP-GS3 proteins were incubated with different concentrations of DTT (dithiothreitol) or $H_2O_2$. As the concentrations of DTT increased, the amount of monomers and dimers gradually increased, and the amount of oligomers gradually decreased. Conversely, when $H_2O_2$ was added, the amount of monomers and dimers gradually decreased, while the amount of oligomers increased as the concentration of $H_2O_2$ were elevated (Fig. 2J), demonstrating that the conformation of GS3 was sensitive to redox environment. Taken together, these results indicated that GS3 could oligomerize through intermolecular disulfide bonds in vitro and in vivo, which is affected by the redox environment.

## The oligomerization of GS3 diminishes its interaction ability with RGB1

It has been reported that GS3 competes with DEP1 or GGC2 to interact with RGB1, thereby restricting grain growth (Sun et al, 2018). We therefore asked whether the oligomerization of GS3 could decrease its interaction with RGB1, promoting grain growth. MBP-GS3 protein was expressed and treated with DTT, $H_2O$ or $H_2O_2$ to induce distinct conformational states. These MBP-GS3 proteins were then incubated with equal amounts of GST-RGB1 protein and MBP beads, and the immunoprecipitates were detected using anti-GST and anti-MBP antibodies, respectively. MBP-GS3 treated with DTT exhibited increased monomers and reduced oligomers, while MBP-GS3 treated with $H_2O_2$ showed decreased monomers and increased oligomers (Fig. 3A). As the oligomerization level of MBP-GS3 increased and the monomer level of

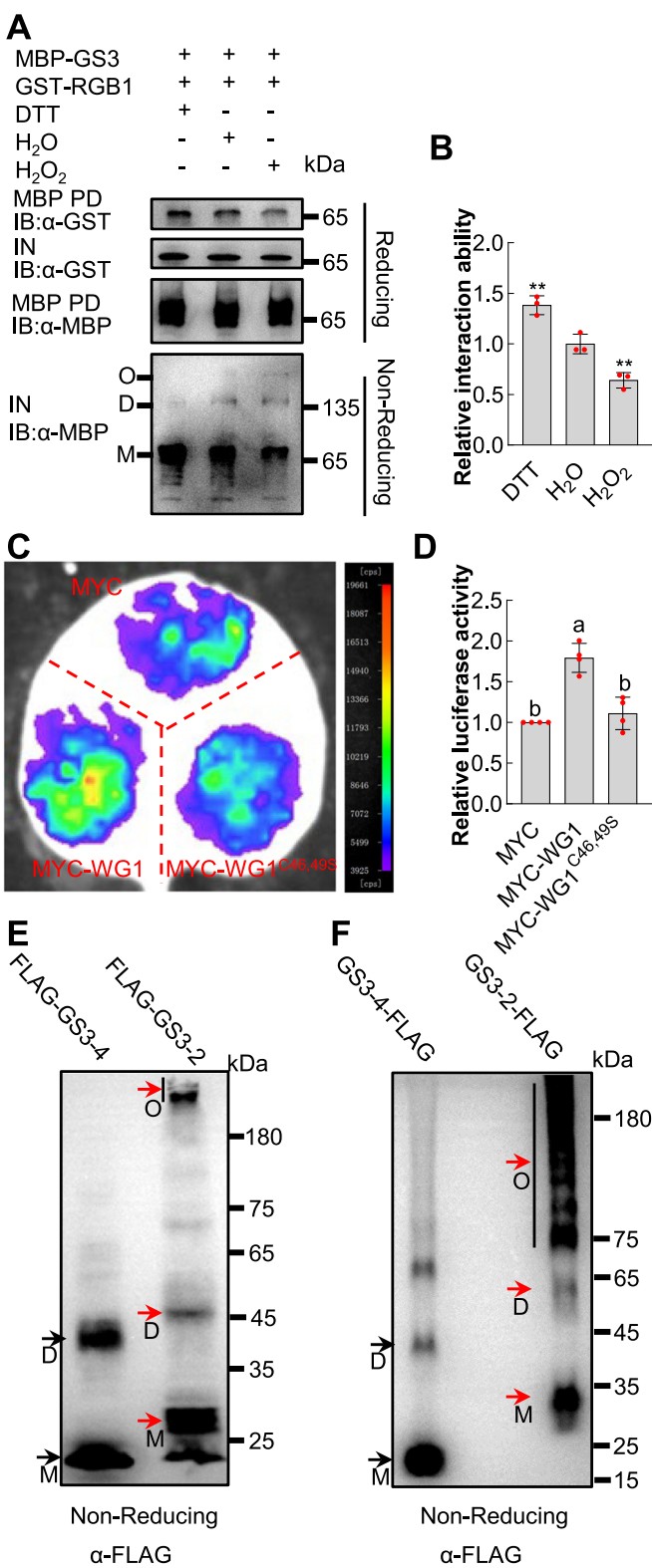

**Figure 3. The oligomerization of GS3 diminishes its interaction ability with RGB1.**

(A) Interaction ability with RGB1 negatively correlates with oligomerization level of GS3 in pull-down assay. MBP-GS3 protein expressed and treated with DTT, $H_2O$ or $H_2O_2$ were incubated with equal amounts of GST-RGB1 protein and MBP beads. The immunoprecipitates were analyzed using anti-GST and anti-MBP antibodies, respectively. The conformational changes of MBP-GS3, treated with DTT, $H_2O$ or $H_2O_2$, were analyzed by separating the proteins on a 10% non-reducing SDS-PAGE gel and then immunoblotting using an anti-MBP antibody. PD, pull-down; IB, immunoblot; IN, input. (B) The relative interaction ability between different oligomerization level of MBP-GS3 and GST-RGB1 from three independent repetitions (A, Fig. EV2A,B) was quantified. The values are means ± SD ($n = 3$). This relative interaction ability was normalized by calculating the ratio of the pulled-down GST-RGB1 to the combined levels of input GST-RGB1 and input MBP-GS3 proteins, with the $H_2O$ group set at 1. The protein level was quantified by ImageJ. The statistical significance was assessed using an unpaired two-tailed t test with Welch's correction in comparison with the $H_2O$ group. ** represents $p \le 0.01$. $p = 0.008$ ($H_2O$ vs. DTT), 0.0085 ($H_2O$ vs. $H_2O_2$). (C) WG1 enhanced the interaction ability between GS3 and RGB1 in the LCI assay. *N. benthamiana* leaves were transformed by injecting *Agrobacterium* GV3101 cells containing *RGB1-nLUC* and *cLUC-GS3* along with *pro35S:MYC*, *pro35S:MYC-WG1* or *pro35S:MYC-WG1^{C46,49S}* plasmids. The luciferase complementation signal was significantly stronger when co-expressed with MYC-WG1 compared to MYC, whereas no significant signal change was observed when co-expressed with MYC-WG1^{C46,49S} compared to MYC. (D) The relative luciferase activity was quantified for each combination, as shown in (C), with the values presented as means ± SD ($n = 4$). The luciferase activity was normalized, with the activity of RGB1-nLUC/cLUC-GS3 co-expressed with MYC set at 1. Different lowercase letters indicate significant differences among the various combinations, as determined by one-way ANOVA with Tukey's multiple comparisons test. $p = 0.0001$ (MYC vs. MYC-WG1), 0.582 (MYC vs. MYC-WG1^{C46,49S}), 0.0004 (MYC-WG1 vs. MYC-WG1^{C46,49S}). (E) FLAG-GS3-4 displayed a weaker oligomerization ability than FLAG-GS3-2 in vitro. FLAG-GS3-4 and FLAG-GS3-2 fusion proteins were expressed, separated on a 10% non-reducing SDS-PAGE gel, and immunoblotted with an anti-FLAG antibody. Monomers and dimers of FLAG-GS3-4 were indicated by black arrows, while monomers, dimers, and oligomers of FLAG-GS3-2 were marked by red arrows. (F) GS3-2-FLAG exhibited a stronger oligomerization ability than GS3-4-FLAG in rice protoplasts. The *pro35S:GS3-2-FLAG* and *pro35S:GS3-4-FLAG* were transiently transformed into rice protoplasts. Subsequently, GS3-2-FLAG and GS3-4-FLAG proteins were expressed and separated on a 10% non-reducing SDS-PAGE gel and immunoblotted with an anti-FLAG antibody. Monomers and dimers of GS3-4-FLAG were indicated by black arrows, while monomers, dimers and oligomers of GS3-2-FLAG were marked by red arrows. M, D, and O in (A), (E), and (F) denote monomers, dimers, and oligomers, respectively. Source data are available online for this figure.

of WG1 on the interaction between GS3 and RGB1 using the LCI assay. Co-expression of MYC-WG1 with cLUC-GS3/RGB1-nLUC resulted in a significant enhancement of luciferase activity compared to the negative control (MYC). By contrast, the mutations of the catalytically active sites of WG1 did not lead to a significant change in luciferase activity compared to the control (MYC) (Fig. 3C,D). These findings supported that the oligomerization status of GS3 influences its interactions with RGB1.

## The C-terminal Cys-rich tail of GS3 is predominantly required for the oligomerization of GS3

There are at least four *GS3* haplotypes according to SNPs in the CDS of *GS3* (Mao et al, 2010). Rice varieties carrying the wild-type GS3-1 or GS3-2 allele, which retain the N-terminal GGL/OSR domain and C-terminal Cys-rich domain (Fig. EV1B), show normal grain length. GS3-2 has a serine insertion compared with the GS3-1 (Fig. EV1B). In this study, *GS3* is the *GS3-2* allele from

MBP-GS3 decreased, the amount of GST-RGB1 pulled down by MBP-GS3 was progressively decreased (Figs. 3A,B and EV2A,B). These findings revealed that the oligomerization of GS3 decreases its interaction ability with RGB1. We further investigated the effect

ZH11 variety. The varieties with the loss-of-function allele GS3-3 form long grains due to the lack of the intact GGL/OSR domain and the C-terminal domain (Mao et al, 2010) (Fig. EV1B). Rice varieties with the gain-of-function GS3-4 allele, which contains the GGL/OSR domain but lacks most of the Cys-rich domain (Fig. EV1A,B), produce very short grains (Mao et al, 2010; Sun et al, 2018). A previous study indicated that GS3-4 has a stronger interaction ability with RGB1 than GS3-1 (Sun et al, 2018). This result prompted us to investigate whether GS3-4 exhibits a weak tendency to oligomerize, thereby increasing its interaction with RGB1 compared with GS3-1 or GS3-2. To test this, FLAG-GS3-4 and FLAG-GS3-2 fusion proteins were expressed in *E. coli* cells, subjected to non-reducing SDS-PAGE, and immunoblotted with an anti-FLAG antibody. As shown in Fig. 3E, FLAG-GS3-2 formed monomers, dimers, and oligomers, while FLAG-GS3-4 predominantly existed in monomeric and dimeric states, indicating that FLAG-GS3-4 has a weaker oligomerization ability than FLAG-GS3-2 in vitro. We further expressed *pro35S:GS3-4-FLAG* and *pro35S:GS3-2-FLAG* in rice protoplasts and investigated their oligomerization levels. Immunoblot assays showed that GS3-2-FLAG formed much stronger oligomerization level than GS3-4-FLAG (Fig. 3F), indicating that the C-terminal Cys-rich tail of GS3 is predominantly required for the oligomerization of GS3 in rice.

## WG1 can reduce the oxidized thiol of GS3

Considering that GS3 can form oligomers through intermolecular disulfide bonds and interact with the functional glutaredoxin WG1, we asked whether WG1 could modify the redox state of GS3. The essence of glutaredoxin-catalyzing-substrate reactions lies in the interaction between the thiol groups within its catalytically active sites and the active thiol groups present in the substrate. During this process, glutaredoxin and its substrate may transiently form an intermediate product (Grx(SH)(SS)(SH)P) linked by a disulfide bond, showcasing a robust protein-protein interaction (Xiao et al, 2019). We therefore investigated whether the catalytically active sites of WG1 is involved in the interaction with GS3. Previous reports indicated that the first and fourth cysteine residues within the catalytic active motif of glutaredoxins may play pivotal roles in catalytic activity on the substrate (Xiao et al, 2019). Based on this, we generated $WG1^{C46,49S}$ with mutations in its catalytically active sites. As shown in Fig. 4A, yeasts transformed with the combination of BD-GS3 and $AD-WG1^{C46,49S}$ grew notably weaker than those transformed with BD-GS3 and AD-WG1 combination on selective medium. Additionally, the relative β-galactosidase activity of the latter combination was significantly stronger than that of the former combination (Fig. 4B). These results suggested that the interaction between GS3 and WG1 is partially reliant on the redox regulation mediated by WG1.

We next tested whether WG1 could reduce the oxidized thiol of GS3 by a biotin-switch assay. MBP-GS3 and FLAG-WG1 fusion proteins were expressed in *E. coli* cells and purified. The purified MBP-GS3 protein was initially treated with $H_2O_2$ to oxidize its reactive thiol groups (-SH). Following this, FLAG-WG1 was introduced to reduce the oxidized residues of MBP-GS3. NEM was then applied to irreversibly block free thiols (-SH). If FLAG-WG1 successfully reduced the oxidized residues (-SOH) of

GS3 back to free thiols, these thiol groups would subsequently be irreversibly blocked by NEM. Subsequently, the application of DTT would reduce the remaining oxidized Cys residues, which can then be detected by an anti-Biotin antibody after BIAM treatment. Therefore, a lower level of BIAM-labeled MBP-GS3 indicates reduction modification by FLAG-WG1 (Fig. 4C). A faint BIAM-labeling band for MBP-GS3 was detected without $H_2O_2$ treatment, suggesting that most active thiols of MBP-GS3 were irreversibly blocked by NEM. When treated with $H_2O_2$, a robust BIAM-labeling band for MBP-GS3 was detected, indicating that $H_2O_2$ could prevent the active thiols of MBP-GS3 from being blocked by NEM. Importantly, FLAG-WG1 weakened BIAM labeling of MBP-GS3, while the catalytically active sites mutation version of FLAG-WG1 ($FLAG-WG1^{C46,49S}$) did not. This emphasized the ability of WG1 to reduce redox-sensitive Cys residues in GS3 (Fig. 4D).

GS3 can oligomerize through intermolecular disulfide bonds in vitro and in vivo, in an oxidized state (Fig. 2H–J). We next investigated whether WG1 could alleviate this oxidized state. Purified MBP-GS3 was incubated with varying amounts of FLAG-WG1 along with 50 mM GSH. The monomer level of MBP-GS3 was significantly increased when FLAG-WG1 was added. Moreover, this increase was further enhanced with the addition of higher concentrations of FLAG-WG1 (Figs. 4E and EV3A,B). By contrast, the incubation with $FLAG-WG1^{C46,49S}$ failed to elicit any alteration in the monomer level of MBP-GS3 (Figs. 4F and EV4A,B). These results supported that WG1 has the ability to reduce the intermolecular disulfide bonds of GS3 in vitro.

We further investigated whether WG1 could reduce the intermolecular disulfide bonds of GS3 in rice plants. To test this, we crossed *pro35S:GFP-GS3* with *pro35S:MYC-WG1* and generated *pro35S:GFP-GS3/pro35S:MYC-WG1* F1 plants. We then compared the conformational changes of the GFP-GS3 protein in *pro35S:GFP-GS3* (homozygous) and *pro35S:GFP-GS3/pro35S:-MYC-WG1* F1 (heterozygous) plants. Total proteins were extracted from young panicles of *pro35S:GFP-GS3* and *pro35S:GFP-GS3/pro35S:MYC-WG1* F1 plants and separated on a non-reducing SDS-PAGE gel. As we expected, total level of GFP-GS3 proteins in *pro35S:GFP-GS3/pro35S:MYC-WG1* F1 plants was lower than that in *pro35S:GFP-GS3* homozygous plants (Fig. 4G). Interestingly, the monomeric form of GFP-GS3 in *pro35S:GFP-GS3/pro35S:MYC-WG1* F1 plants was obviously more than that in *pro35S:GFP-GS3* homozygous plants (Fig. 4G). Furthermore, we transiently transformed *35S:FLAG-GS3* into wild-type and *wg1-2* protoplasts. We then compared the conformational changes of the FLAG-GS3 protein in wild-type and *wg1-2* protoplasts. Under non-reducing conditions, FLAG-GS3 proteins were detected in an oligomeric state (Fig. EV5). Upon the addition of a little β-ME (partial reducing), the oligomerization level of FLAG-GS3 was significantly higher in the *wg1-2* mutant than that in the wild type (Fig. 4H). Together, these results demonstrated that WG1 can reduce the intermolecular disulfide bonds of GS3 in rice.

## *WG1* acts genetically with *GS3* to regulate grain length

Given that WG1 physically interacts with GS3 and influences its redox state, we sought to understand the genetic relationship between *GS3* and *WG1* in grain size control. To test this, we

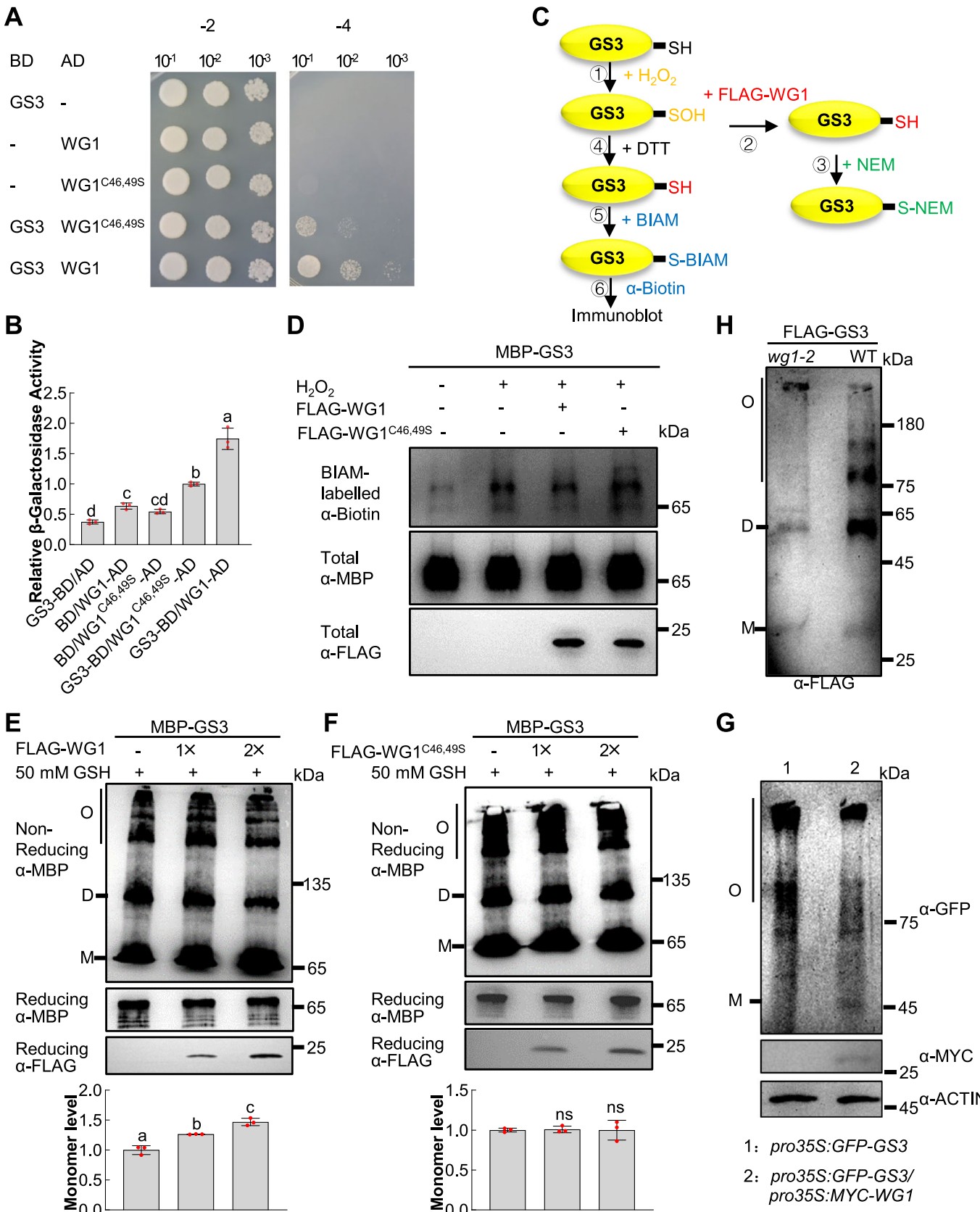

**Figure 4.  WG1 can reduce the oxidized thiol of GS3.**

(A) The catalytically active site CCMC of WG1 is crucial to its interaction with GS3. The indicated construct pairs were co-transformed into yeast strain AH109. Interactions between bait and prey were examined on the control media SD-2 (SD/-Leu/-Trp) and selective media SD-4 (SD/-Leu/-Trp/-His/-Ade). AD-WG1 and BD pair, BD-GS3 and AD pair as well as AD-WG1$^{C46,49S}$ and BD pair served as negative controls. The notation "C46, 49S" indicates that the 46th and 49th cysteine residues of WG1 protein have been mutated to serine, resulting in a catalytically inactive version of WG1. (B) The relative β-galactosidase activity was quantified for each pair of bait and prey proteins as indicated in (A). The values are means ± SD ($n = 3$). The average value of GS3-BD/WG1$^{C46,49S}$-AD pair was set at 1. Different lowercase letters denote significant differences among the various pairs, as determined by one-way ANOVA with Tukey's multiple comparisons test. $p = 0.0242$ (GS3-BD/AD vs. BD/WG1-AD), 0.1794 (GS3-BD/AD vs. BD/WG1$^{C46,49S}$-AD), 0.6906 (BD/WG1-AD vs. BD/WG1$^{C46,49S}$-AD), 0.0028 (BD/WG1-AD vs. GS3-BD/WG1$^{C46,49S}$-AD), 0.0005 (BD/WG1$^{C46,49S}$-AD vs. GS3-BD/WG1$^{C46,49S}$-AD), <0.0001 for other comparisons. (C) Flow diagram of biotin-switch assay for detecting whether WG1 could reduce oxidized thiol of GS3. Purified MBP-GS3 were treated with or without H$_2$O$_2$ (①), then incubated with or without FLAG-WG1/FLAG-WG1$^{C46,49S}$ (②) and followed by reacting with NEM (③), DTT (④), and BIAM (⑤). The BIAM labeled MBP-GS3 were detected using an anti-Biotin antibody (⑥). (D) Oxidized thiol of GS3 can be reduced by WG1 in biotin-switch assay. Light BIAM-labeling bands for MBP-GS3 without H$_2$O$_2$ treatment indicated that most active thiols of MBP-GS3 were blocked by NEM. Strong BIAM-labeling bands for MBP-GS3 with H$_2$O$_2$ treatment indicated that H$_2$O$_2$ can protect the active thiol of MBP-GS3 from being blocked by NEM. This protective effect was significantly decreased with FLAG-WG1 treatment, whereas it was not decreased by the treatment with the catalytically active site mutation version of FLAG-WG1 (FLAG-WG1$^{C46,49S}$), indicating the reduction of MBP-GS3 oxidized thiol by FLAG-WG1. The inputs of MBP-GS3 and FLAG-WG1 were detected using an anti-MBP and anti-FLAG antibodies, respectively. All proteins were separated on 10% reducing SDS–PAGE gels for analysis. (E) WG1 can reduce the intermolecular disulfide bonds of GS3 in vitro. Purified MBP-GS3 proteins, incubated with or without varying concentrations of FLAG-WG1 in the presence of 50 mM GSH, were separated on a 6% non-reducing SDS–PAGE gel and immunoblotted using an anti-MBP antibody. The inputs of MBP-GS3 and FLAG-WG1 were separated on a 10% reducing SDS–PAGE gel and immunoblotted using an anti-MBP and anti-FLAG antibodies, respectively. With an increase in the concentration of FLAG-WG1, the monomers of MBP-GS3 showed a gradual increase obviously. The protein level of MBP-GS3 monomers/inputs MBP-GS3 ratio, representing the monomer level of MBP-GS3 from three independent repeats (**E**, Fig. EV3A,B), was illustrated in the bar chart below, with the treatment lacking FLAG-WG1 set at 1. The protein level was quantified by ImageJ. The values are means ± SD ($n = 3$). Different lowercase letters denote significant differences among the various groups, as determined by one-way ANOVA with Tukey's multiple comparisons test. $p = 0.0026$ (CK vs. 1 X FLAG-WG1), 0.0001 (CK vs. 2 X FLAG-WG1), 0.0094 (1 X FLAG-WG1 vs. 2 X FLAG-WG1). (F) FLAG-WG1$^{C46,49S}$ loses its ability to reduce the intermolecular disulfide bonds of GS3 in vitro. Similar experimental procedures were conducted as presented in (**E**), with FLAG-WG1 replaced with FLAG-WG1$^{C46,49S}$. After mutating the catalytically active sites CCMC of WG1 to an inactive SCMS form, the oligomerization level of MBP-GS3 could no longer be reduced. The protein level of MBP-GS3 monomers/inputs MBP-GS3 ratio, representing the monomer level of MBP-GS3 from three independent repeats (**F**, Fig. EV4A,B), was illustrated in the bar chart below, with the treatment lacking FLAG-WG1$^{C46,49S}$ set at 1. The protein level was quantified by ImageJ. The values are means ± SD ($n = 3$). The $p$ value was determined using an unpaired two-tailed t test with Welch's correction compared with the absence of FLAG-WG1$^{C46,49S}$ treatment. ns represents no significant difference. $p = 0.7346$ (CK vs. 1 X FLAG-WG1$^{C46,49S}$), 0.9887 (CK vs. 2 X FLAG-WG1$^{C46,49S}$). (G) MYC-WG1 diminishes the oligomerization level of GFP-GS3 in rice. Total proteins extracted from 10 to 15 cm young panicles of *pro35S:GFP-GS3* (homozygous, 1) and *pro35S:GFP-GS3/pro35S:MYC-WG1* F1 (heterozygous, 2) plants were separated on a 10% non-reducing SDS-PAGE gel and immunoblotted using an anti-GFP antibody. The inputs of MYC-WG1 and ACTIN were separated on 10% reducing SDS–PAGE gels and immunoblotted using anti-MYC and anti-ACTIN antibodies, respectively. (H) FLAG-GS3 exhibited a stronger oligomerization ability in the *wg1-2* mutant compared to the wild type in rice protoplasts. The *pro35S:FLAG-GS3* vector was transiently transformed into both *wg1-2* and wild-type protoplasts. FLAG-GS3 proteins were expressed and separated on a 10% partial-reducing (4% β-ME treatment) SDS-PAGE gel and immunoblotted with an anti-FLAG antibody. M, D, and O in (**E–H**) denote monomers, dimers, and oligomers, respectively. Source data are available online for this figure.

generated a loss-of-function mutant *gs3-c1* in the ZH11 background using CRISPR/Cas9 technology (Fig. 5A). The *gs3-c1* mutant had a 1-bp deletion in the first exon of *GS3*, leading to a reading frame shift in the GGL domain and a premature stop codon (Figs. 5A,B and EV1B). The *gs3-c1* grains were significantly longer than those of ZH11 (Fig. 5C,E), consistent with previous studies (Fan et al, 2006; Mao et al, 2010). Meanwhile, we generated *proActin:WG1* transgenic lines in ZH11 background. Consistent with a previous study (Hao et al, 2021), the *proActin:WG1#1* transgenic line produced wider but shorter grains than ZH11 (Fig. 5C,E,F). We crossed *gs3-c1* with *proActin:WG1#1* and generated *proActin:WG1#1;gs3-c1* plants for genetic analyses. We first examined the expression levels of *WG1* in *proActin:WG1#1;gs3-c1* and *proActin:WG1#1* and found that they had similar *WG1* expression levels (Fig. 5D). We then investigated the grain size phenotypes of ZH11, *proActin:WG1#1, gs3-c1* and *proActin:WG1#1;gs3-c1*. The grain length of *proActin:WG1#1* was decreased by 2.8% compared to that of ZH11, whereas the length of *proActin:WG1#1;gs3-c1* grains was similar to that of *gs3-c1* grains (Fig. 5C,E). The grain width of *proActin:WG1#1;gs3-c1* was similar to that of *proActin:WG1#1* (Fig. 5C,F), possibly because *gs3-c1* did not significantly influence grain width (He et al, 2024) (Fig. 5C,F). These findings supported that WG1 and GS3 function in a common pathway to regulate grain length.

## Disscussion

Grain size is an important yield trait in crops. Rice GS3, an atypical Gγ subunit, is a key regulator of grain length, and its loss-of-function allele has been widely used for grain length improvement in elite varieties (Zeng et al, 2019; Zhou et al, 2019). Despite its importance, it remains unclear how GS3 activity is regulated to control grain length. In this study, we discover that GS3 can form dimers and oligomers by intermolecular disulfide bonds, which repress its function. The CC-type glutaredoxin WG1 interacts with GS3 and reduces the oligomerizaiton level of GS3, thereby promoting the function of GS3 (Fig. 6). Our findings reveal a previously unrecognized mechanism that redox regulation of GS3 by the glutaredoxin WG1 controls grain length, opening a perspective for G protein signaling regulation.

Glutaredoxins play a crucial role in maintaining redox homeostasis (Chai and Mieyal, 2023; Sevilla et al, 2023). We have previously demonstrated that a functional glutaredoxin WG1 negatively controls grain length but positively regulates grain width (Hao et al, 2021). However, it remains unknown whether WG1 can modulate redox state of its substrates to regulate grain size. Here we demonstrated that WG1 regulates the redox state of GS3 to control grain length. We discovered that GS3 can form dimers and oligomers via intermolecular disulfide bonds (Fig. 2H–J). WG1 physically interacts with GS3 in the cytoplasm

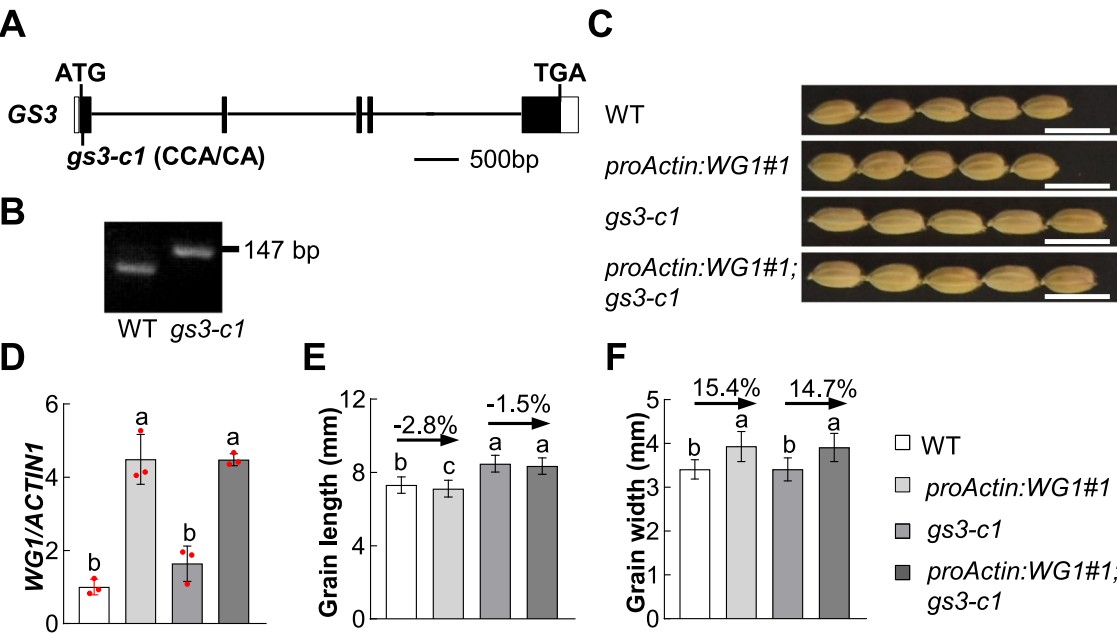

**Figure 5.** *GS3* acts in a common pathway with *WG1* to regulate grain length.

(A) The gene structure of *GS3*. The start codon (ATG) and the stop codon (TGA) are indicated. Black boxes indicate the CDS, and white boxes show the 5'- and 3'-untranslated regions. The mutation site of *gs3-c1* is shown. The *gs3-c1* was generated by CRISPR/Cas9-based genome editing. It has a 1-bp deletion compared to the wild type ZH11 sequence. (B) The dCAPS marker is developed according to the *gs3-c1* mutation. The restriction enzyme *Bgl* I was used to digest the PCR product. (C) Grains of wild type, *proActin:WG1#1*, *gs3-c1* and *proActin:WG1#1;gs3-c1* plants. Bars = 1 cm. (D) Relative quantitative real-time PCR analysis of *WG1* expression in 10-day old seedlings of wild type, *proActin:WG1#1*, *gs3-c1* and *proActin:WG1#1;gs3-c1* plants. (E, F) Grain length (E) and grain width (F) of wild type, *proActin:WG1#1*, *gs3-c1* and *proActin:WG1#1;gs3-c1* plants. Values in (D) to (F) are means ± SD ($n = 3$ for **D**, $n = 117/135/127/94$ for **E** and **F**). Different lowercase letters denote significant differences among the various groups, as determined by one-way ANOVA with Tukey's multiple comparisons test. $p = 0.3488$ (WT vs. *gs3-c1*), 0.0002 (*proActin:WG1#1* vs. *gs3-c1*), 0.0002 (*gs3-c1* vs. *proActin:WG1#1;gs3-c1*), <0.0001 (WT vs. *proActin:WG1#1*), <0.0001 (WT vs. *proActin:WG1#1;gs3-c1*), >0.9999 (*proActin:WG1#1* vs. *proActin:WG1#1;gs3-c1*) in (**D**). $p = 0.003$ (WT vs. *proActin:WG1#1*), 0.1835 (*gs3-c1* vs. *proActin:WG1#1;gs3-c1*), <0.0001 for other comparisons in (**E**). $p = 0.9387$ (*proActin:WG1#1* vs. *proActin:WG1#1;gs3-c1*), >0.9999 (WT vs. *gs3-c1*), <0.0001 for other comparisons in (**F**). Source data are available online for this figure.

adjacent to the plasma membrane and other parts of the cytoplasm (Fig. 1D). WG1 reduces the oligomerization of GS3 by reducing intermolecular disulfide bonds (Fig. 4E–H). Consistent with this, WG1 predominantly exists in a reduced state (as monomers) in the cytoplasm (Appendix Fig. S1). Previous studies proposed that the interaction of RGB1 with GGC2/DEP1 promotes grain growth in rice. GS3 can inhibit the interaction between RGB1 and GGC2/DEP1 by competitively binding RGB1, thereby restricting grain growth (Sun et al, 2018). Here we revealed that the increased oligomerization level of GS3 decreases its interaction with RGB1, while the decreased oligomerization level of GS3 increases its interaction with RGB1 (Fig. 3A,B). Therefore, the oligomerization of GS3 may reduce its function due to a decrease in the competitive binding to RGB1 (Fig. 6). Given that appropriate level of ROS is necessary for plant growth and development (Huang et al, 2019; Mittler, 2017), it is possible that ROS signal may influence the oligomerization of GS3 in rice plants. By contrast, it seems reasonable that WG1 promotes the function of GS3 in grain length control by reducing the oligomerization of GS3 through redox regulation (Fig. 6). Consistent with this, our genetic analyses showed that *gs3-c1* mutant suppresses the short grain phenotype of *proActin:WG1#1* (Fig. 5C,E), supporting that the short grain phenotype of *proActin:WG1#1* depends on the functional GS3. By contrast, the grain width of *proActin:WG1#1;gs3-c1* was similar to

that of *proActin:WG1#1* (Fig. 5C,F), possibly because *gs3-c1* did not obviously influence grain width (Fig. 5C,F). This also suggested that WG1 may regulate grain width by influencing other substrates. Therefore, the redox regulation of GS3 by WG1 is crucial for grain length control in rice. It is possible that redox modulation of G proteins may represent a crucial regulatory mechanism of G protein signaling in plants and animals, although it remains to be fully investigated in the future.

The Cys-rich tail of GS3 plays a mysterious role in grain length control (Mao et al, 2010). Varieties harboring the gain-of-function allele GS3-4, which retains the GGL/OSR domain and loses most of the Cys-rich tail (Fig. EV1A), form the very short grains (Mao et al, 2010; Sun et al, 2018). It has been proposed that the Cys-rich tail may have an inhibitory effect on the GGL/OSR domain (Sun et al, 2018). GS3-4 has stronger inhibitory effect on the interaction of RGB1 with DEP1/GGC2 than GS3-1 (Sun et al, 2018). However, it still remains unclear why the Cys-rich tail influences the function of GS3. In this study, we found that the oligomerization of GS3-4 was dramatically decreased compared with that of GS3-2, highlighting the significant role of the C-terminal Cys-rich sequence in the oligomerization of GS3. Thus, it is plausible that the dramatically decreased oligomerization of GS3-4 may strongly repress the interaction of RGB1 with GGC2/DEP1 by competitively binding RGB1, thereby resulting in a gain-of-function effect. These results

can interpret a long standing question why the C-terminal Cys-rich tail of GS3 negatively influences its function. These findings also provide an opportunity to precisely improve grain length by finely adjusting the length of Cys-rich tail of GS3 in the future.

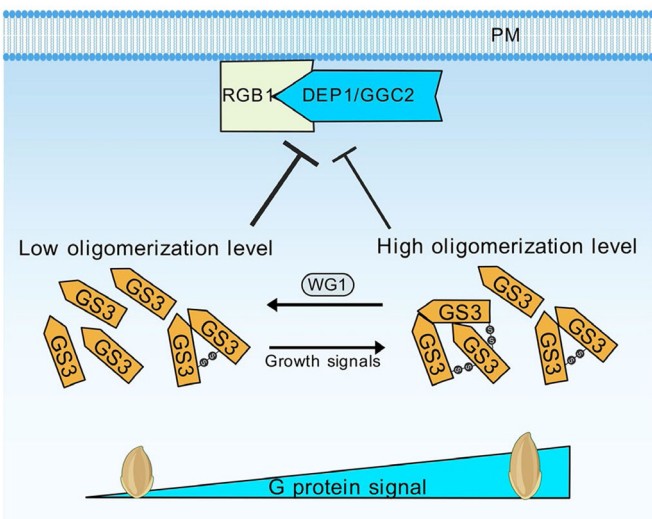

**Figure 6. A model for WG1-mediated redox regulation of G protein signaling in grain length control.**

The interaction between RGB1 and DEP1/GGC2 promotes grain growth in the grain-length direction. GS3 can form dimers and oligomers via intermolecular disulfide bonds when receiving growth signals. The oligomerization of GS3 diminishes its inhibitory effect on the interaction between RGB1 and DEP1/GGC2, thereby promoting grain growth in the grain-length direction. Glutaredoxin WG1 has the ability to reduce the oligomerization of GS3. This, in turn, promotes the inhibitory effect of GS3 on the interaction between RGB1 and DEP1/GGC2, thereby decreasing grain growth in the grain-length direction. PM plasma membrane. The figure was created using the Generic Diagramming Platform available at https://biogdp.com/ (Jiang et al, 2025).

## Methods

### Reagents and tools table

| Reagent/Resource | Reference or Source | Identifier or Catalog Number |
|---|---|---|
| **Experimental models** | | |
| gs3-c1 | This study | |
| pro35S:MYC-WG1 | This study | |
| pro35S:GFP-GS3 | This study | |
| proActin:WG1 | This study | |
| proWG1:WG1-GFP | (Hao et al, 2021) | |
| wg1-2 | (Hao et al, 2021) | |
| BL21 Competent Cells | Biomed | BC201 |
| GV3101 Electrocompetent Cells | Biomed | BC308 |
| **Recombinant DNA** | | |
| pro35S:MYC-WG1 | This study | |
| pro35S:GFP-GS3 | This study | |

| Reagent/Resource | Reference or Source | Identifier or Catalog Number |
|---|---|---|
| proActin:WG1 | (Hao et al, 2021) | |
| pCAMBIA1300-cas9-GS3N | This study | |
| AD-WG1 | (Hao et al, 2021) | |
| BD-WG1 | (Hao et al, 2021) | |
| AD-GS3 | This study | |
| AD-GS3^1-94 | This study | |
| AD-GS3^95-232 | This study | |
| AD-WG1^C46,49S | This study | |
| BD-GS3 | This study | |
| cYFP-GS3 | This study | |
| nYFP-GS3 | This study | |
| nYFP-WG1 | This study | |
| PIP2-mCherry | (Yang et al, 2021b) | |
| H2B-mCherry | Lijun Luo's lab from Shanghai Agrobiological Gene Center | |
| VPS23A-mCherry | (Yang et al, 2021b) | |
| MBP-GS3 | This study | |
| GST-RGB1 | This study | |
| FLAG-WG1 | (Hao et al, 2021) | |
| cLUC-GS3 | This study | |
| cLUC-EOG1 | This study | |
| RGB1-nLUC | This study | |
| GS3-nLUC | This study | |
| WG1-nLUC | (Hao et al, 2021) | |
| EOG1-nLUC | (Yan et al, 2024) | |
| MYC-WG1^C46,49S | This study | |
| MBP-FLAG | This study | |
| FLAG-WG1^C46,49S | This study | |
| FLAG-GS3-4 | This study | |
| FLAG-GS3-2 | This study | |
| GS3-4-FLAG | This study | |
| GS3-2-FLAG | This study | |
| FLAG-GS3 | This study | |
| **Antibodies** | | |
| Anti-MBP | New England Biolabs | E8032S |
| Anti-GST | Abmart | MA9025 |
| Anti-GFP | Abmart | M20004 |
| Anti-H4 | Active Motif | 61521 |
| Anti-BiP | Agrisera | AS09-481 |
| Anti-ACTIN | EASYBIO | BE0027-100 |
| Anti-FLAG | Abmart | M20008M |
| Anti-MYC | Abmart | M20002 |
| Anti-MYC | Cell Signaling TECHNOLOGY | 2276S |
| Anti-Biotin | Cell Signaling TECHNOLOGY | 5597S |

| Reagent/Resource | Reference or Source | Identifier or Catalog Number |
|---|---|---|
| **Oligonucleotides and other sequence-based reagents** | | |
| Primers | This study | Appendix Table S1 |
| **Chemicals, Enzymes and other reagents** | | |
| *Bam*H I | New England Biolabs | R3136S |
| *Pac* I | New England Biolabs | R0547L |
| *Asc* I | New England Biolabs | R0558S |
| T4 DNA ligase | New England Biolabs | M0202S |
| *Aar* I | Thermo Fisher Scientific | ER1581 |
| *Kpn* I | New England Biolabs | R3142L |
| *Bgl* II | New England Biolabs | R0144L |
| *Eco*R I | New England Biolabs | R3101S |
| *Sac* I | New England Biolabs | R3156L |
| *Pst* I | New England Biolabs | R0140S |
| *Nco* I | New England Biolabs | R3193L |
| *Xba* I | New England Biolabs | R0145L |
| *Sal* I | New England Biolabs | R3138S |
| *Not* I | New England Biolabs | R3189S |
| *Hind* III | New England Biolabs | R3104L |
| *Xho* I | New England Biolabs | R0146L |
| *Bst*B I | New England Biolabs | R0519V |
| *Spe* I | New England Biolabs | R3133S |
| *Bgl* I | New England Biolabs | R0143S |
| MES | GPCSCI | AN443 |
| MgCl$_2$ | Sigma | M2393 |
| Acetosyringone | Macklin | A800901 |
| Isopropyl β-d-1thiogalactopyranoside | Amresco | 0487 |
| HEPES | NOVON SCIENTIFIC | MC0753 |
| Triton X-100 | Sigma | T9284 |
| Glycerol | Sigma | G5516 |
| NaCl | Sigma | V900058 |
| EGTA | Amresco | 0732 |
| Protease inhibitor cocktail | Roche | 04693132001 |
| d-luciferin | Promega | E1602 |
| NEM | Sigma-Aldrich | E3876 |
| SDS loading buffer (non-Reducing) | CWBIO | 0028S |
| H$_2$O$_2$ | Sinopharm Chemical Reagent | 10011208 |
| DTT | Amresco | 0281 |
| β-Mercaptoethanol | Amresco | M131 |
| GSH | Sigma | G4251 |
| Acetone | Sinopharm Chemical Reagent | 10000418 |
| BIAM | Anaspec | AS-60644 |
| SYBR qPCR Mix | TOYOBO | QKD-201 |

| Reagent/Resource | Reference or Source | Identifier or Catalog Number |
|---|---|---|
| MBP beads | New England Biolabs | E8021S |
| GFP-Trap® Agarose beads | Chromo Tek | AB_2631357 |
| **Software** | | |
| ImageJ | https://imagej.net/ | Version 1.54f |
| GraphPad Prism | https://www.graphpad.com | Version 10.1.2 |
| SC-G | Wseen | Version 1.3.7.0 |
| **Other** | | |
| RNA Extraction Kit | ZOMANBIO | ZP405K-2 |
| cDNA Synthesis Kit | Vazyme | R212-02 |
| ClonExpress II One Step Cloning Kit | Vazyme | C112-1 |
| β-Galactosidase Activity Assay Kit | Coolaber | YK3030 |
| Camera | Nikon | D7200 |
| qRT-PCR machine | Eppendorf | Realplex2 |
| qRT-PCR machine | Roche | Lightcycler 480 |
| Confocal laser scanning microscope | Zeiss | LSM980 |
| Gel-imaging system | Tanon | 4500 |
| In vivo imaging system | Berthold | NightOWL II LB 983 |
| Sonicator | Qsonica | Q700 |
| Scanner | Microtek | Scan Marker i560 |
| Luminometer | Promega | GloMax 96 |
| Generic Diagramming Platform | https://biogdp.com/ | |

## Plant materials and growth conditions

Rice *gs3-c1* mutant was generated by CRISPR-Cas9 technology in the background of Zhonghua 11 (ZH11). All of the rice plants were grown either in a greenhouse at 30 °C/25 °C (day/night) or in the standard experimental fields in Changping (116°20′E, 40°22′N, Beijing, China) from May to October or Lingshui (110°03′E, 18°51′N, Hainan province, China) from December to the April of the following year under natural conditions.

## Morphological analysis

Morphological analyses were performed according to previous descriptions (Huang et al, 2017). Rice grains were harvested after reaching full ripeness. Next, the grains were scanned using Scan Marker i560 (Microtek, China). We then measured grain width and length using the SC-G software (Wseen, China). For every sample, we measured more than 50 grains to calculate the mean values of grain length and width using the GraphPad Prism software (version 10.1.2). We selected 5 grains with average grain length and width to photograph using a Nikon D7200 camera.

## RNA extraction, reverse transcription, and quantitative real-time PCR (qRT-PCR) assays

Total RNA from rice seedlings was extracted through an RNA extraction kit (ZOMANBIO, ZP405K-2). cDNAs were synthesized by reverse transcription assay using a reverse transcription kit (Vazyme, R212-02). All of the above assays were performed following the manufacturer's instructions. qRT-PCR was conducted on an Eppendorf Mastercycler ep Realplex2 or Lightcycler 480 (Roche) instrument using a SYBR qPCR Mix (KOD, QKD-201). Rice *ACTIN1* was used to normalize expression. Primers used for qRT-PCR assays are listed in the Appendix Table S1.

## Plasmid construction and plant transformation

All plants expression vectors were constructed using a ClonExpress II One Step Cloning Kit (Vazyme, C112-1). Primers used for amplifying coding sequences (CDSs) are listed in Appendix Table S1. The CDS of *WG1* and *GS3* were amplified from the cDNA of ZH11 and then inserted into the linearized *pCAMBIA1300-221-Myc* (Liu et al, 2011) (digested with *Bam*H I and *Pac* I) and *pMDC43* (digested with *Asc* I and *Pac* I) vector to generate *pro35S:MYC-WG1* and *pro35S:GFP-GS3*, respectively. The *proActin:WG1* vector were constructed in our previous work (Hao et al, 2021).

The CRISPR-Cas9 vectors were constructed using a T4 DNA ligase (NEB, M0202S). Target sequences were contained in the primers listed in the Appendix Table S1. To generate the *pCAMBIA1300-cas9-GS3N* vector, we first inserted the annealed primers contained the target sequences to the *SK-gRNA* vector (Wang et al, 2015) (digested with *Aar* I) using T4-DNA ligase to generate *SK-gRNA-GS3N*. Then, we digested *SK-gRNA-GS3N* using *Kpn* I *and Bgl* II to get the expression cassettes. Finally, the expression cassettes were inserted into the *Bam*H I and *Kpn* I sites of the *pCAMBIA1300-cas9* vector to generate the *pCAMBIA1300-cas9-GS3N* vector. All of the above vectors were transferred into *Agrobacterium* strain GV3101 (Biomed, BC308) and subsequently transformed into ZH11 as described previously (Hiei et al, 1994).

## Yeast two-hybrid (Y2H) assays

We performed the Y2H assay with Matchmaker Gold Yeast Two-Hybrid System from Clontech. *AD-WG1* and *BD-WG1* vectors used for Y2H assay were described in previous reports (Hao et al, 2021). The CDSs of *GS3*, *GS3$^{1-94}$* (amino acids [aa]: 1–94), *GS3$^{95-232}$* (aa: 95–232), *WG1$^{C46,49S}$* were all amplified from ZH11 cDNA. Specially, the CDS of *WG1$^{C46,49S}$* was generated using an overlap extension PCR method. Primers used for amplification were listed in the Appendix Table S1. Then, we cloned those CDSs into the *Eco*R I and *Sac* I sites of pGADT7 to generate *AD-GS3*, *AD-GS3$^{1-94}$*, *AD-GS3$^{95-232}$* and *AD-WG1$^{C46,49S}$*, respectively. The CDS of *GS3* was amplified using primers (listed in the Appendix Table S1) and cloned into the *pGBKT7* (double digested with *Pst* I and *Nco* I) to generate *BD-GS3*. Different combinations of these vectors were transformed into yeast strain AH109 following the manufacturer's instructions. Quantitation of β-galactosidase (β-gal) activity was determined as described by the manufacturer's instructions (Coolaber, YK3030).

## Bimolecular fluorescence complementation (BiFC) assays

For BiFC assay in *N. benthamiana*, the CDSs of *GS3* and *WG1* were amplified from cDNA of ZH11, and the N-terminal and C-terminal fragments of YFP (nYFP and cYFP) were amplified from *cYFP-SMG3* and *nYFP-DGS1* vectors, which had been generated in our previous work (Li et al, 2023). We used overlap extension PCR to generate fused DNA fragment *Fu- cYFP-GS3*, *Fu- nYFP-GS3* and *Fu- nYFP-WG1*. Primers used for CDSs amplification are listed in the Appendix Table S1. Next, we cloned those three fused DNA fragments into linearized vector *pGBW414* (double digested with *Xba* I and *Sal* I) to generate *cYFP-GS3*, *nYFP-GS3* and *nYFP-WG1*, respectively. We then transformed those vectors and subcellular markers (PIP2-mCherry, H2B-mCherry, VPS23A-mCherry) into *Agrobacterium* GV3101 cells. Various combinations of *Agrobacterium* cells, resuspended in an activation buffer (10 mM MES/KOH pH 5.6, 10 mM MgCl$_2$ and 150 μM acetosyringone), were injected into leaves of *N. benthamiana* plants. After 36–72 h, we detected the YFP and mCherry fluorescence using LSM980 confocal laser scanning microscope (Zeiss).

## Pull-down assays

To generate prokaryotic expression vectors *MBP-GS3* and *GST-RGB1*, we inserted the CDSs of *GS3* and *RGB1* into the linearized *pMAL-c2* (double digested by *Eco*R I and *Pst* I) and *pGEX-4T-1* (double digested by *Bam*H I and *Not* I) vectors, respectively. Primers used for CDSs amplification are listed in the Appendix Table S1. The *FLAG-WG1* vector had been generated in our previous work (Hao et al, 2021). Then, those vectors were all transformed into *Escherichia coli* (*E. coli*) BL21 cells. The corresponding proteins were then expressed in these cells by adding isopropyl β-d-1-thiogalactopyranoside (IPTG) to a final concentration of 0.4 mM and incubating at 28 °C for 3–4 h. Then, the cells were sonicated in ice-cold TGH buffer (50 mM HEPES pH 7.5, 1% Triton X-100, 10% glycerol, 1.5 mM MgCl$_2$, 150 mM NaCl, 1 mM EGTA and 1 tablet of EDTA-free protease inhibitor cocktail (Roche) per 50 ml buffer) to release the proteins into the buffer, and the supernatant was collected by centrifugation at 4 °C for use in pull-down assays. Next, equal amounts of the corresponding proteins were added to 1 ml of ice-cold TGH buffer, followed by the addition of 25 μl MBP beads (amylase resin, New England Biolabs, E8021S). Subsequently, all the samples were incubated at 4 °C for 45–60 min with gentle shaking. Finally, the MBP beads were washed at least 5 times using ice-cold TGH buffer, followed by the addition of 50 μl of 2× SDS loading buffer and boiling at 98 °C for at least 10 min. The denatured proteins were separated using 10% or 15% SDS–PAGE gels. We detected the corresponding proteins using an anti-MBP antibody (NEB, E8032S; dilution, 1:10,000), anti-GST antibody (Abmart, MA9025; dilution, 1:5000) or anti-FLAG (Abmart, M20008; dilution, 1:5000). A Tanon-4500 gel-imaging system were used to scan the images.

## Firefly luciferase complementation imaging (LCI) assay

The CDSs of *GS3, EOG1,* and *RGB1* were inserted into the linearized *pCAMBIAsplit_cLUC* (double digested by *Bam*H I and *Sal* I) and *pCAMBIAsplit_nLUC* (double digested by *Bam*H I and

*Sal* I) to generate *cLUC-GS3, cLUC-EOG1, RGB1-nLUC,* and *GS3-nLUC*, respectively. The *WG1-nLUC* and *EOG1-nLUC* was generated in our previous work (Hao et al, 2021; Yan et al, 2024). To construct *MYC-WG1^{C46,49S}* vector, we amplified the CDS of *WG1^{C46,49S}* from *AD-WG1^{C46,49S}* and inserted it into *pCAMBIA1300-221-Myc* (double digested by *Kpn* I and *Bam*H I). Primers used for amplification are listed in the Appendix Table S1. All of those vectors were transformed into *Agrobacterium* GV3101 cells. Different combinations of GV3101 cells were resuspended in an activation buffer and then infiltrated into leaves of *N. benthamiana* to express the corresponding proteins in darkness for 36–48 h. To detect the luciferase activities, we sprayed 1 mM d-luciferin solution (Promega, E1602) onto the *N. benthamiana* leaves and allowed it to react for 2 min. A NightOWL II LB 983 system was used to capture images (Chen et al, 2008) and the luciferase activity was tested using 96 microplate luminometer instrument from Promega.

## Subcellular fractionation and immunoblot assay

We used the *proWG1:WG1-GFP* stable transgene plants, generated previously (Hao et al, 2021), to conduct subcellular fractionation assay following the methods described earlier (Huang et al, 2017). We used the luminal-binding protein BiP as the marker protein for the cytoplasmic protein fraction (C) and histone H4 as the marker protein for the nuclear protein fraction (N). WG1-GFP, histone H4, and BiP were detected using antibodies anti-GFP (Abmart, M20004; dilution, 1:5000), anti-H4 (Active Motif, 61521; dilution, 1:500), and anti-BiP (Agrisera, AS09-481; dilution, 1:2000), respectively.

## Redox reagent treatment assay and protein oligomerization level detection

For in vitro NEM treatment assay, MBP-FLAG was used as a negative control. *MBP-FLAG* was constructed by annealing and ligation method. The annealed primers were inserted into linearized *pMAL-c2* (double-digested by *Bam*H I and *Hind* III) using T4 DNA ligase. Primers used for annealing are listed in Appendix Table S1. BL21 cells transformed with *MBP-GS3* or *MBP-FLAG* were resuspended using ice-cold RS buffer (50 mM HEPES pH 7.5, 10% glycerol, 1% Triton X-100, 150 mM NaCl and 1 tablet of EDTA-free protease inhibitor cocktail (Roche) per 50 ml buffer) with NEM (Sigma, E3876) additional added to a final concentration of 100 mM, then sonicated for 1 min (1 s on and 10 s off) at 20 amplitudes using a sonicator (Qsonica, Q700). Then, 40 μl of MBP beads were added to the supernatant and incubated at a 4 °C cold room for 45–60 min with gentle rotation. Next, we washed the MBP beads at least 5 times using ice-cold Pur-wash buffer (50 mM HEPES pH 7.5, 10% glycerol and 150 mM NaCl) with NEM additional added to a final concentration of 50 mM. Finally, the purified MBP-GS3 or MBP-FLAG were eluted by 50 μl MBP elution buffer (200 mM NaCl, 20 mM Tris-HCl pH 7.4, 10 mM maltose and 10% glycerol). The purified MBP-GS3 and MBP-FLAG were then mixed with 30 μl 1× non-reducing SDS loading buffer (CWBIO, 0028S) and boiled at 98 °C for at least 5 min. The purification process for the MBP-GS3 and MBP-FLAG protein without NEM treatment followed the same steps as described above, with NEM excluded at each stage. The separated

proteins were detected by western blot with an anti-MBP antibody (NEB, E8032S; dilution, 1:10,000).

For in vitro redox reagents treatment assay, equal amounts of purified MBP-GS3 proteins were treated with or without 1 mM or 0.5 mM of $H_2O_2$ or 1 mM, 2 mM, 5 mM, or 8 mM of DTT at 25 °C in dark for 5 min with gentle shaking. Proteins were detected as described above.

For in vivo redox reagents treatment assay, we used young panicles of *pro35S:GFP-GS3* stable transgenic plants to extract total proteins using an ice-cold plant protein extraction buffer (150 mM NaCl, 2% TritonX-100, 50 mM Tris-HCl pH 7.4, 20% glycerol, 1 mM EDTA and 1 tablet protease inhibitor cocktail (Roche) per 50 ml buffer). The extracted total proteins were then divided into two equal volumes and treated with or without 30% β-Mercaptoethanol. Total proteins were separated by non-reducing SDS-PAGE. The GFP-GS3 proteins were detected using an anti-GFP antibody (Abmart, M20004; dilution, 1:5000). An antibody (EASYBIO, BE0027-100; dilution, 1:5000) against plant ACTIN was used to verify the equal loading of GFP-GS3 proteins.

For in vitro detection of the oligomerization level of MBP-GS3, we first generated *FLAG-WG1^{C46,49S}* vector by insert the CDS of *WG1^{C46,49S}*, amplified from *AD-WG1^{C46,49S}*, into the linearized *pETNT* vector (digested by *Eco*R I and *Xho* I). Primers used for CDS amplification were listed in Appendix Table S1. Then, equal amounts of purified MBP-GS3 proteins were incubated with different amounts of FLAG-WG1 or FLAG-WG1^{C46,49S} in a room temperature with gental shaking for 45 min. The input MBP-GS3 and FLAG-WG1 or FLAG-WG1^{C46,49S} were detected using an anti-MBP antibody (NEB, E8032S; dilution, 1:10,000) and an antibody against FLAG (Abmart, M20008M; dilution, 1:5000), respectively. A non-reducing SDS-PAGE was used to detect the oligomerization stature of MBP-GS3. Proteins were detected as mentioned above.

To detect the oligomerization level of GFP-GS3 in vivo, total proteins were extracted from young panicles of *pro35S:GFP-GS3* homozyzous plants and *pro35S:GFP-GS3/pro35S:MYC-WG1* F1 plants using a plant protein extraction buffer mentioned above supplemented with an additional 2% GSH (Sigma, G4251). The corresponding extracted total proteins were then added 2× non-reducing SDS loading buffer and boiled at 98 °C for 10 min before SDS-PAGE analysis. The GFP-GS3 were detected as described above. An anti-MYC antibody (Abmart, M20002; dilution, 1:5000) were used to detect MYC-WG1. ACTIN was used as a loading control and detected as mentioned above.

For in vitro detection of the oligomerization level of FLAG-GS3-4 and FLAG-GS3-2, we used overlap extension PCR to amplify the CDS of *GS3-4* from the cDNA of ZH11, which is also used in a previous report (Ferrero-Serrano et al, 2024). The CDS of *GS3-2* were also amplified from cDNA of ZH11. We then inserted the CDS of *GS3-4* and *GS3-2* into the linearized *pETNT* vector (double digested by *Eco*R I and *Xho* I) to generate *FLAG-GS3-4* and *FLAG-GS3-2*, respectively. The two vectors were subsequently transformed into *E. coli* BL21 cells to express the corresponding protein under the previously described conditions. Total proteins were separated using non-reducing SDS-PAGE. An antibody against FLAG (Abmart, M20008M; dilution, 1:5000) were used to detect the FLAG-GS3-4 and FLAG-GS3-2 protein.

To detect the in vivo oligomerization levels of GS3-4-FLAG and GS3-2-FLAG, we amplified the CDSs of *GS3-4* and *GS3-2* from vectors *FLAG-GS3-4* and *FLAG-GS3-2*, respectively. Then, the

amplified CDSs were cloned into *Kpn* I and *Bst*B I sites of the *pUC-FLAG* vector to generate *GS3-4-FLAG* and *GS3-2-FLAG*, respectively. We then transformed those two vectors into the rice protoplasts prepared from young seedlings of ZH11. The corresponding proteins were separated by a non-reducing SDS-PAGE. Proteins were detected using an antibody against FLAG (Abmart, M20008M; dilution, 1:5000).

To compare the conformation of FLAG-GS3 in *wg1-2* and the wild type, we initially inserted the CDS of *GS3*, amplified from ZH11 cDNA, into the *Spe* I and *Bst*B I sites of *FLAG-pUC* to generate the *FLAG-GS3* vector. Primers are listed in Appendix Table S1. Subsequently, we transformed it into the protoplasts generated from young seedlings of both *wg1-2* and the wild type. An antibody against FLAG (Abmart, M20008M; dilution, 1:5000) was used to detect proteins.

### BIAM labeling assay

BIAM labeling assay was performed as described before (Tian et al, 2018; Yang et al, 2021a). In detail, purified MBP-GS3 and MBP-FLAG were first incubated with 0, 10, and 1000 μM $H_2O_2$ at a room temperature in darkness for 15 min in a BIAM labeling buffer (50 mM MES-NaOH, pH 6.5, 100 mM NaCl and 1% TritonX-100). Then, no more than four volumes of acetone (pre-cold) was added to precipitate proteins in −20 °C for 20–30 min and centrifuged at $5000 \times g$ for 7–10 min in 4 °C. The pellets were washed for three times with cold acetone (no more than 80%) and dissolved in a BIAM labeling buffer contained 500 μM BIAM (Anaspec, AS-60644), then reacted at a room temperature with gentle shaking in darkness for 1 h. The reaction was stopped by the adding final concentration of 20 mM β-mercaptoethanol. Proteins were precipitated by cold acetone as described above. The pellets were dissolved in 50 μl SDS loading buffer and separated using a SDS-PAGE. An antibody against Biotin (Cell Signaling TECHNOLOGY, Cat: 5597S; dilution, 1:1000) was used to detect the proteins labeled with BIAM. Total input MBP-FLAG and MBP-GS3 were detected with an antibody against MBP (NEB, Cat: E8032S; dilution, 1:10,000).

### Biotin-switch assay

The biotin-switch assay was used, with slight modifications to the previously described methods (Tian et al, 2018; Yang et al, 2021a), to show the redox status of MBP-GS3. The recombinant MBP-GS3 protein was purified from *E. coli*, then treated with or without of 200 μM $H_2O_2$ at a room temperature in darkness for 15 min. Proteins were precipitated using cold acetone as described in the BIAM labeling assay. The treated proteins were then incubated with purified FLAG-WG1 or FLAG-WG1$^{C46,49S}$, or without either of these at a room temperature with gentle shaking under darkness for 1 h. Next, the following reagents were added to the reaction system in order: NEM, DTT, and BIAM. Meanwhile, after each treatment step, proteins were precipitated and washed three times with cold acetone (no more than 80%) to prevent reagent interference. BIAM labeled proteins were detected with Rabbit anti-Biotin (Cell Signaling TECHNOLOGY, 5597S; dilution, 1:1000). The input MBP-GS3 was detected using an antibody against MBP (NEB, E8032S; dilution, 1:10,000). FLAG-WG1 and FLAG-WG1$^{C46,49S}$

were detected using an antibody against FLAG (Abmart, M20008M; dilution, 1:5000).

### In vivo co-immunoprecipitation

Total proteins from young panicles (10–15 cm) of *pro35S:GFP-GS3;pro35S:MYC-WG1* and *pro35S:GFP;pro35S:MYC-WG1* plants were extracted using ice-cold protein extraction buffer mentioned above and incubated with 40 μl GFP-Trap® Agarose beads (Chromo Tek, AB_2631357) for 1 h with gentle shaking in 4 °C cold room. Then, the GFP-Trap® Agarose beads were washed at least three times using ice-cold CO-IP wash buffer (50 mM Tris pH 7.5, 0.1% TritonX-100, 10% glycerol, 150 mM NaCl, 1 mM EDTA and 1 tablet protease inhibitor cocktail (Roche) per 50 ml buffer). The immunoprecipitated and input proteins were analyzed using SDS-PAGE and immunoblotted by anti-GFP (Abmart, M20004; dilution, 1:5000) and anti-MYC (Cell Signaling TECHNOLOGY, 2276S; dilution, 1:1000) antibodies.

## Data availability

This study includes no data deposited in external repositories. The authors declare that the data supporting the findings of this study are available within the article or are available from the corresponding authors upon reasonable request.

The source data of this paper are collected in the following database record: biostudies:S-SCDT-10_1038-S44318-025-00462-9.

## Peer review information

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

## Acknowledgements

We would like to thank Dr. Li Yan for help with the *cLUC-EOG1* vector. We thank Professors Mingyi Bai and Yanchen Tian from Shandong University for sharing the protocols for the BIAM-labeling assay and Biotin-switch assay. We thank Professors Kang Chong and Yunyuan Xu from University of Chinese Academy of Sciences for *PIP2-mCherry* and *VPS23A-mCherry* plasmids. We thank Professor Lijun Luo from Shanghai Agrobiological Gene Center for *H2B-mCherry* plasmid. This work is supported by grants from Biological Breeding-National Science and Technology Major Project (2023ZD0406802-JQH), the National Natural Science Foundation of China (32301852-JQH), the National Key Research and Development Program of China (2022YFF1002903-YHL and 2021YFF1000202-YHL), the Hainan Seed Industry Laboratory (B21HJ0220-YHL), the Key Research and Development Program of Hainan (ZDYF2021XDNY165-YHL) and the strategic priority research program of the Chinese Academy of Sciences (XDB1090000-YHL).

## Author contributions

**Lijie Liu**: Resources; Data curation; Formal analysis; Validation; Investigation; Visualization; Methodology; Writing—original draft; Writing—review and editing. **Jianqin Hao**: Resources; Data curation; Formal analysis; Funding acquisition; Validation; Investigation; Visualization; Methodology; Writing—original draft; Writing—review and editing. **Ke Huang**: Resources; Methodology. **Penggen Duan**: Resources. **Baolan Zhang**: Resources. **Zhihai Chi**: Methodology. **Xiaohong Yao**: Resources. **Yunhai Li**: Conceptualization; Supervision; Funding acquisition; Project administration; Writing—review and editing.

Source data underlying figure panels in this paper may have individual authorship assigned. Where available, figure panel/source data authorship is listed in the following database record: biostudies:S-SCDT-10_1038-S44318-025-00462-9.

## Disclosure and competing interests statement

The authors declare no competing interests.

# Expanded View Figures

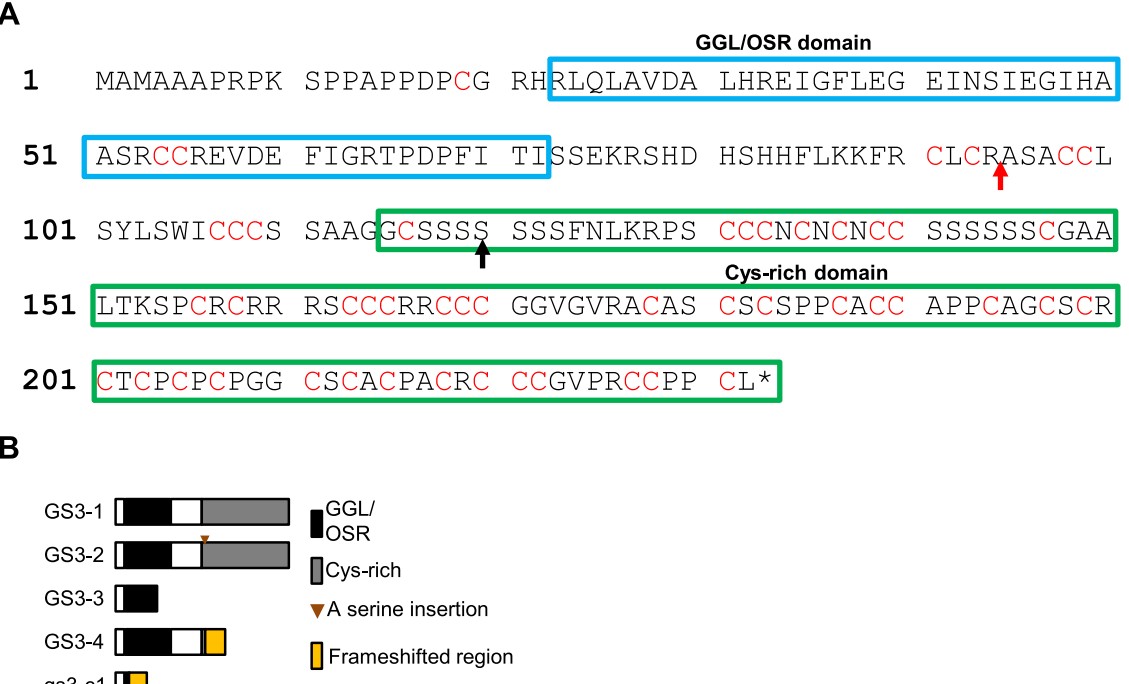

**Figure EV1.  Amino acids sequence of GS3 protein in ZH11 variety and protein structures of different haplotypes of *GS3*.**

(**A**) The amino acid sequence of GS3 from the ZH11 variety, which belongs to the *GS3-2* haplotype, is shown. The GGL/OSR and Cys-rich domains are highlighted in blue and green boxes, respectively, with cysteines labeled in red. The Cys-rich domain comprises 33.9% cysteines (40 out of 118 amino acids), while the C-terminal of GS3 in Fig. 2C, truncated by a red arrow, contains 32.6% cysteines (45 out of 138 amino acids). Additionally, the protein of the *GS3-4* haplotype was frameshifted at the site indicated by the black arrow. (**B**) The protein structures of different haplotypes of *GS3* and *gs3-c1* mutant are shown. Compared to haplotype GS3-1, GS3-2 has a serine insertion without any other variations. GS3-3 encodes a truncated GGL/OSR domain. GS3-4 preserves the GGL/OSR domain but loses most of the Cys-rich domain. gs3-c1 retains a small portion of the GGL/OSR domain and contains a frameshifted region.

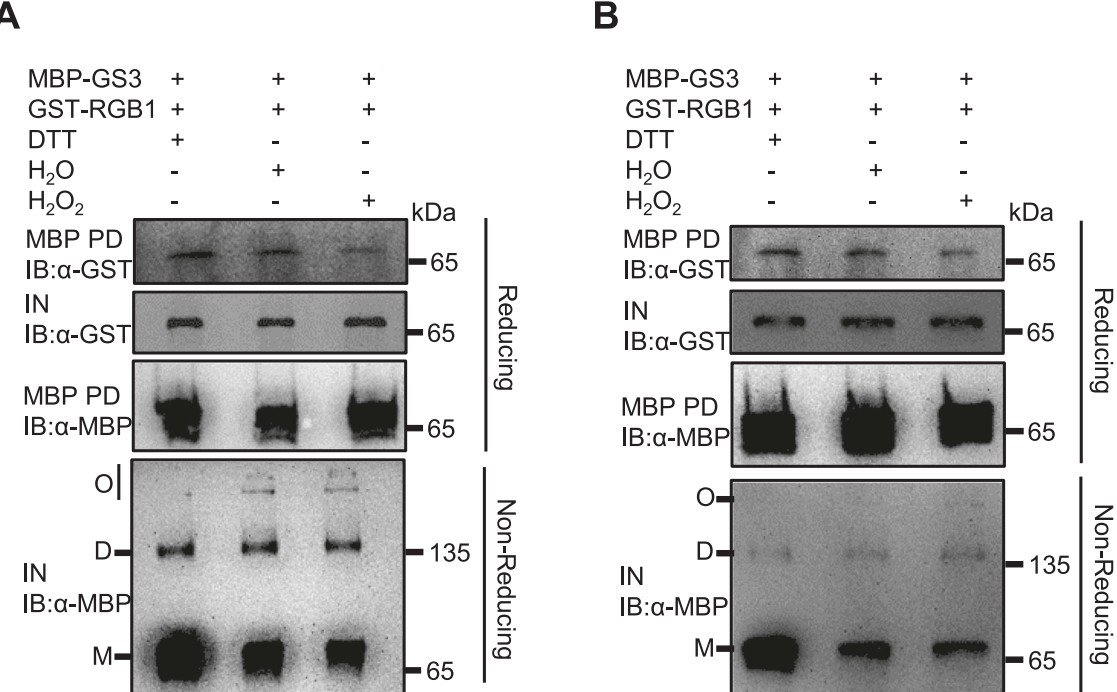

**Figure EV2. The oligomerization of GS3 diminishes its interaction ability with RGB1.**

(A, B) Additional two independent repeat experiments of Fig. 3A are shown. M, D, and O in (A, B) denote monomers, dimers, and oligomers, respectively.

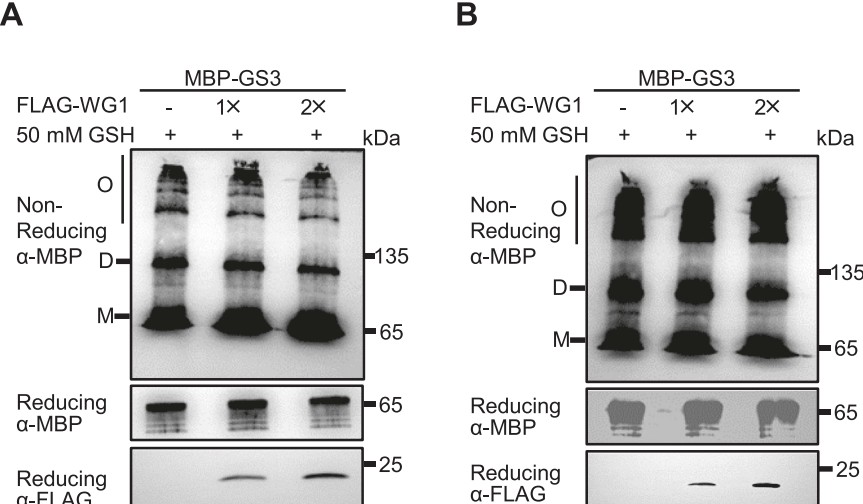

**Figure EV3. WG1 can reduce the intermolecular disulfide bonds of GS3 in vitro.**

(A, B) Additional two independent repeat experiments of Fig. 4E are shown. M, D, and O in (A, B) denote monomers, dimers, and oligomers, respectively.

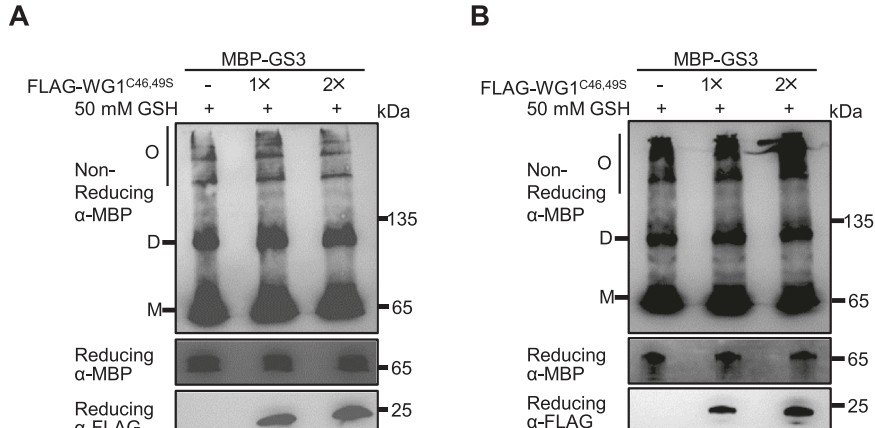

**Figure EV4.  FLAG-WG1^(C46,49S) loses its ability to reduce the intermolecular disulfide bonds of GS3 in vitro.**

(A, B) Additional two independent repeat experiments of Fig. 4F are shown. M, D, and O in (A, B) denote monomers, dimers, and oligomers, respectively.

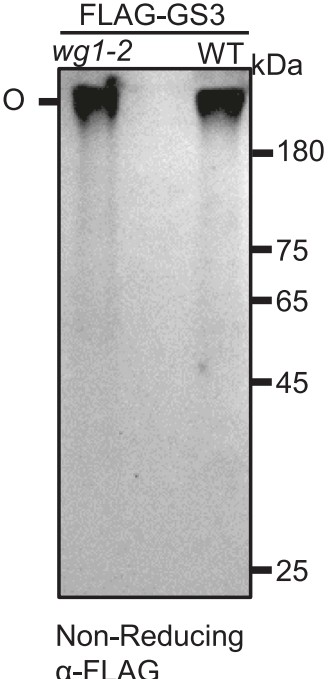

**Figure EV5.   FLAG-GS3 proteins were detected in an oligomeric state in both *wg1-2* and wild-type protoplasts under non-reducing conditions.**

The *pro35S:FLAG-GS3* vector was transiently transformed into both *wg1-2* and wild-type protoplasts. FLAG-GS3 proteins were expressed and separated on a 10% non-reducing SDS-PAGE gel and immunoblotted with an anti-FLAG antibody. O denotes oligomers.

