## [Peer Review File · The EMBO Journal]

Redox regulation of G protein oligomerization and signaling by the glutaredoxin WG1 controls grain size in rice

Lijie Liu, Jianqin Hao, Ke Huang, Penggen Duan, Baolan Zhang, Zhihai Chi, Xiaohong Yao, and Yunhai Li *Corresponding author(s): Yunhai Li (yhli@genetics.ac.cn)*

Review Timeline:

Submission Date:	1st Sep 24
Editorial Decision:	7th Oct 24
Revision Received:	5th Mar 25
Editorial Decision:	28th Mar 25
Revision Received:	8th Apr 25
Accepted:	24th Apr 25

Editor: William Teale

Transaction Report:

Dear Dr. Li,

Thank you again for the submission of your manuscript entitled "Redox regulation of G protein signaling by WG1 controls grain size in rice" and for your patience during the review process. We have now received the reports from the referees, which I copy below.

As you can see from their comments, while all referees appreciated the relevance and quality of the work you present, Referee #2 and Referee #3 request that your results be put into a more robust tissue and cellular context. This (and some requested control experiments) will require your attention before your manuscript can be published in The EMBO Journal.

Based on the overall interest expressed in the reports, I would like to invite you to address the comments of all referees in a revised version of the manuscript. I should add that it is The EMBO Journal policy to allow only a single major round of revision and that it is therefore important to resolve the main concerns at this stage. I believe the concerns of the referees are reasonable and addressable, but please contact me if you have any questions, need further input on the referee comments or if you anticipate any problems in addressing any of their points. I am available via Zoom to discuss the reports with you at any time; let me know if you would like to go through them together. Please, follow the instructions below when preparing your manuscript for resubmission.

I would also like to point out that as a matter of policy, competing manuscripts published during this period will not be taken into consideration in our assessment of the novelty presented by your study ("scooping" protection). We have extended this 'scooping protection policy' beyond the usual 3 month revision timeline to cover the period required for a full revision to address the essential experimental issues. Please contact me if you see a paper with related content published elsewhere to discuss the appropriate course of action.

Again, please contact me at any time during revision if you need any help or have further questions.

Thank you very much again for the opportunity to consider your work for publication. I look forward to your revision.

Best regards,

William

William Teale, Ph.D.
Editor
The EMBO Journal

When submitting your revised manuscript, please carefully review the instructions below and include the following items:

- 1) a .docx formatted version of the manuscript text (including legends for main figures, EV figures and tables). Please make sure that the changes are highlighted to be clearly visible.
- 2) individual production quality figure files as .eps, .tif, .jpg (one file per figure).
- 3) a .docx formatted letter INCLUDING the reviewers' reports and your detailed point-by-point response to their comments. As part of the EMBO Press transparent editorial process, the point-by-point response is part of the Review Process File (RPF), which will be published alongside your paper.
- 4) a complete author checklist, which you can download from our author guidelines ([https://wol-prod-cdn.literatumonline.com/pb-assets/embo-site/Author Checklist%20-%20EMBO%20J-1561436015657.xlsx](https://wol-prod-cdn.literatumonline.com/pb-assets/embo-site/Author%20Checklist%20-%20EMBO%20J-1561436015657.xlsx)). Please insert information in the checklist that is also reflected in the manuscript. The completed author checklist will also be part of the RPF.
- 5) Please note that all corresponding authors are required to supply an ORCID ID for their name upon submission of a revised manuscript.
- 6) We require a 'Data Availability' section after the Materials and Methods. Before submitting your revision, primary datasets produced in this study need to be deposited in an appropriate public database, and the accession numbers and database listed

under 'Data Availability'. Please remember to provide a reviewer password if the datasets are not yet public (see <https://www.embopress.org/page/journal/14602075/authorguide#datadeposition>). If no data deposition in external databases is needed for this paper, please then state in this section: This study includes no data deposited in external repositories. Note that the Data Availability Section is restricted to new primary data that are part of this study.

Note - All links should resolve to a page where the data can be accessed.

8) For data quantification: please specify the name of the statistical test used to generate error bars and P values, the number (n) of independent experiments (specify technical or biological replicates) underlying each data point and the test used to calculate p-values in each figure legend. The figure legends should contain a basic description of n, P and the test applied. Graphs must include a description of the bars and the error bars (s.d., s.e.m.).

9) We would also encourage you to include the source data for figure panels that show essential data. Numerical data can be provided as individual .xls or .csv files (including a tab describing the data). For 'blots' or microscopy, uncropped images should be submitted (using a zip archive or a single pdf per main figure if multiple images need to be supplied for one panel). Additional information on source data and instruction on how to label the files are available at .

10) We replaced Supplementary Information with Expanded View (EV) Figures and Tables that are collapsible/expandable online (see examples in <https://www.embopress.org/doi/10.15252/embj.201695874>). A maximum of 5 EV Figures can be typeset. EV Figures should be cited as 'Figure EV1, Figure EV2" etc. in the text and their respective legends should be included in the main text after the legends of regular figures.

12) Our journal encourages inclusion of *data citations in the reference list* to directly cite datasets that were re-used and obtained from public databases. Data citations in the article text are distinct from normal bibliographical citations and should directly link to the database records from which the data can be accessed. In the main text, data citations are formatted as follows: "Data ref: Smith et al, 2001" or "Data ref: NCBI Sequence Read Archive PRJNA342805, 2017". In the Reference list, data citations must be labeled with "[DATASET]". A data reference must provide the database name, accession number/identifiers and a resolvable link to the landing page from which the data can be accessed at the end of the reference. Further instructions are available at .

13) In order to increase the reproducibility and reach of your work, The EMBO Journal includes a table of reagents that were used in the study. Please provide this along with your revisions.

Further instructions for preparing your revised manuscript:

- a point-by-point response to the referees' comments, with a detailed description of the changes made (as a word file).
- a word file of the manuscript text.
- individual production quality figure files (one file per figure)
- a complete author checklist, which you can download from our author guidelines (<https://www.embopress.org/page/journal/14602075/authorguide>).

- Expanded View files (replacing Supplementary Information)

We realize that it is difficult to revise to a specific deadline. In the interest of protecting the conceptual advance provided by the work, we recommend a revision within 3 months (5th Jan 2025). Please discuss the revision progress ahead of this time with the editor if you require more time to complete the revisions. Use the link below to submit your revision:

Referee #1:

The findings reported in this manuscript build the previous studies of the authors, who provided evidence that the glutaredoxin WG1 negatively controls rice grain length but positively regulates grain width (Hao et al., 2021). The approaches and methodology used are appropriate and the results are convincing. The study is significant because data are presented showing that WG1 regulates the redox state of GS3, in a manner that alters the oligomerization of the protein. Crucially, WG1 is shown to physically interact with GS3 at the plasma membrane and in the cytoplasm. This interaction is dependent on the GS3 oligomerization level, increased oligomerisation decreasing the interaction with RGB1. Evidence for the formation of GS3 dimers and oligomers via redox modulation of intermolecular disulfide bonds is presented. These changes involve the redox modulation of the cysteine (Cys)-rich C-terminus of GS3. The data are appropriately presented and discussed. I am particularly impressed by the clarity of the images. Taken together, these data provide a realistic and novel mechanism for the redox regulation grain length in rice.

I have no major concerns regarding the data and conclusions, and hence I see no reason why the work cannot be published as it stands. I consider that this paper presents new and interesting information and hence it will be well cited.

Referee #2:

G protein signaling is a conserved pathway that regulates grain/seed size in rice and Arabidopsis. Of this pathway, GS3 acts as a central regulator for grain size regulation in rice, and its loss-of-function allele has been widely utilized by breeders to improve appearance quality by modifying grain length. Dissecting the fine regulatory network of GS3 and identifying new factors for this pathway will provide opportunities to coordinate improvements in grain yield and quality. In this manuscript, building upon previous findings that WG1 exhibits disulfide oxidoreductase activity and functions in grain size, Liu and his/her colleagues further identified GS3 as the redox substrate of WG1, and thus expand our understanding of grain length regulation network in rice. Overall, the authors provided strong molecular, biochemical, and genetic evidence to support their main conclusions. However, some experiments require additional controls, and partial conclusions need more solid data for robust support. Below are my concerns:

Major concerns:

1. In Figure 1D, the authors mentioned that the YFP fluorescence was observed in both plasma membrane and cytoplasm. However, considering that WG1-GFP was only detected in the cytoplasm and nucleus (Figure 1E and Hao et al., 2021, Molecular Plant), the observation of "YFP fluorescence in the plasma membrane from BiFC" remains unclear. It is suggested to add both plasma membrane and cytoplasm markers in the BiFC experiments or rephrase this statement.
2. When comparing the interaction strength of proteins (Figure 2C and 4A), the authors should provide additional results beyond yeast growth, such as β -galactosidase assay or luciferase complementation assay (LCI).
3. In Figures 1C and Figure 2D, the authors performed LCI assay to test the interactions of GS3 and WG1 and with itself. While the authors included the negative control with empty vector nLUC and cLUC, it would be beneficial to verify whether nLUC and cLUC are translated into proteins in these empty controls or include an unrelated protein with the same subcellular location as suggested in the publication: "Zhou, Z., Bi, G., & Zhou, J.-M. (2018). Luciferase complementation assay for protein-protein interactions in plants. *Current Protocols in Plant Biology*, 3, 42-50. doi: 10.1002/cppb.20066".
4. The authors claimed that the oligomerization level of GS3 regulates its interaction with RGB1 when treated with DTT, H₂O and H₂O₂ in Figures 3A and 3B. I wonder how the interaction between GS3 and RGB1 is influenced when WG1 is co-expressed in Yeast Three-Hybrid, LCI or pulldown assays.
5. In Figure 4F, the authors stated that MYC-WG1 reduces the oligomerization level of GFP-GS3 in rice. Is the oligomerization of GS3 increased in the *wg* mutant (Hao et al., 2021, Molecular Plant)?
6. As the authors mentioned, the loss-of-function allele of GS3 has been widely utilized by breeders to improve grain length, but the effect of improving grain length varies among different cultivars. Therefore, I am curious about the natural variation of WG1 among different rice cultivars, landraces and wild rice.

Minor concerns:

1. In Line 84-87, the authors cited the same publication in different places, but used different indexes. It would be better to maintain consistency in the citation format.
2. WD1 showed weak interaction with GS3 during the seedling stage (Figure 1E-F). According to previous reports, WG1 is expressed at its highest during YP1 stage. Therefore, is it possible to conduct a co-immunoprecipitation (CO-IP) assay for GS3 and WG1 during YP1 stage?
3. In Line 128, *N. benthamiana* should be spelled out when used first.
4. Line 195, The authors claimed that "Co-expression of Cyfp-GS3 with nYFP-GS3 resulted in strong YFP fluorescence in the plasma membrane and the cytoplasm of epidermal cells in *N. benthamiana* leaves". However, the strong fluorescence dots were observed in Figure 2E. What is the nature of these dots?
5. In Line 334, WG1 and GS3 should be italic.

Referee #3:

Grain size is crucial for crop yield. The CC-type GRX protein WG1/OsGRX8 has been demonstrated to have disulfide oxidoreductase activity and be involved in regulation of grain size in rice. However, the specific substrate that WG1 targets to influence grain length remains unclear. In this study, Liu et al., identified GS3, a protein that controls grain length in rice, as an interacting partner of WG1. The authors demonstrated that WG1 physically interacts with GS3 to regulate grain length. They also discovered that GS3 can form oligomers through intermolecular disulfide bonds, which reduce its ability to interact with RGB. Furthermore, WG1 enhances GS3's function in grain length control by decreasing its oligomerization. These findings contribute to our understanding of yield regulation and suggest that WG1 could serve as a molecular target for breeding high-yield rice. However, further experiments are needed to fully validate the authors' conclusions.

1. Both WG1 and GS3 are localized in the nucleus and cytoplasm. However, the authors observed the interaction between WG1 and GS3 only in the plasma membrane and cytoplasm. Furthermore, WG1 is expressed more strongly in the nucleus than in the cytoplasm. Did the authors compare the state of WG1 protein between the nucleus and cytoplasm? This comparison could be significant for understanding its function
2. The authors conducted a Co-IP assay to demonstrate the interaction between WG1 and GS3 in transgenic lines, extracting total proteins from 10-day-old seedlings. However, they should provide Co-IP assays from young panicles, as the interaction between WG1 and GS3 is relevant to grain length
3. The authors conducted BIAM labeling assays to demonstrate that GS3 can oligomerize through intermolecular disulfide bonds (Figure 2B). However, this experiment lacks proper controls, which is also a concern in Figures 2F and 4C.
4. The authors indicated that the oligomerization of GS3 weakens its interaction with RGB1. They should provide additional experiments to support this claim.

Dear Reviewers,

Thank you very much for your supportive and helpful suggestions on the manuscript (EMBOJ-2024-118898). As you suggested, we conducted new experiments to address the questions you raised. These experiments yielded new results that further supported our main conclusion and strengthened the story. They have helped make a better paper. We would like to submit the revision for possible publication in EMBO Journal.

Referee #1:

The findings reported in this manuscript build the previous studies of the authors, who provided evidence that the glutaredoxin WG1 negatively controls rice grain length but positively regulates grain width (Hao et al., 2021). The approaches and methodology used are appropriate and the results are convincing. The study is significant because data are presented showing that WG1 regulates the redox state of GS3, in a manner that alters the oligomerization of the protein. Crucially, WG1 is shown to physically interact with GS3 at the plasma membrane and in the cytoplasm. This interaction is dependent on the GS3 oligomerization level, increased oligomerisation decreasing the interaction with RGB1. Evidence for the formation of GS3 dimers and oligomers via redox modulation of intermolecular disulfide bonds is presented. These changes involve the redox modulation of the cysteine (Cys)-rich C-terminus of GS3. The data are appropriately presented and discussed. I am particularly impressed by the clarity of the images. Taken together, these data provide a realistic and novel mechanism for the redox regulation grain length in rice.

I have no major concerns regarding the data and conclusions, and hence I see no reason why the work cannot be published as it stands. I consider that this paper presents new and interesting information and hence it will be well cited.

Response: Thanks for your very positive comments.

Referee #2:

G protein signaling is a conserved pathway that regulates grain/seed size in rice and Arabidopsis. Of this pathway, GS3 acts as a central regulator for grain size regulation in rice, and its loss-of-function allele has been widely utilized by breeders to improve appearance quality by modifying grain length. Dissecting the fine regulatory network of GS3 and identifying new factors for this pathway will provide opportunities to coordinate improvements in grain yield and quality. In this manuscript, building upon previous findings that WG1 exhibits disulfide oxidoreductase activity and functions in grain size, Liu and his/her colleagues further identified GS3 as the redox substrate of WG1, and thus expand our understanding of grain length regulation network in rice. Overall, the authors provided strong molecular, biochemical, and genetic evidence to support their main conclusions.

However, some experiments require additional controls, and partial conclusions need more solid data for robust support. Below are my concerns:

Response: Thanks for your positive comments and suggestions. We have also carried out additional experiments according to your suggestions, and results further strengthened our main conclusion as detailed below.

Major concerns:

1. In Figure 1D, the authors mentioned that the YFP fluorescence was observed in both plasma membrane and cytoplasm. However, considering that WG1-GFP was only detected in the cytoplasm and nucleus (Figure 1E and Hao et al., 2021, Molecular Plant), the observation of "YFP fluorescence in the plasma membrane from BiFC" remains unclear. It is suggested to add both plasma membrane and cytoplasm markers in the BiFC experiments or rephrase this statement.

Response: Thanks for your helpful suggestions. As you suggested, we further assessed the co-localization of the interaction site with a plasma membrane marker (PIP2-mCherry) in the BiFC assay. The signals were found to co-localize with the

plasma membrane marker (PIP2-mCherry), and signals were also observed in the inner region of the cell (referred to as the cytoplasm) (Figure 1D). Because WG1 is localized in both the cytoplasm and the nucleus (Appendix Figure S1), WG1 associates with GS3 in the cytoplasm adjacent to the plasma membrane as well as other parts of the cytoplasm. We rephrased this in this revision (Line132-149).

2. When comparing the interaction strength of proteins (Figure 2C and 4A), the authors should provide additional results beyond yeast growth, such as β -galactosidase assay or luciferase complementation assay (LCI).

Response: As you suggested, we repeated the interaction assay in yeast cells, further assessed β -galactosidase activity for each bait-prey protein pair, and observed that weaker interactions corresponded to lower β -galactosidase activity (Figure 2C-E and Figure 4A-B)

3. In Figures 1C and Figure 2D, the authors performed LCI assay to test the interactions of GS3 and WG1 and with itself. While the authors included the negative control with empty vector nLUC and cLUC, it would be beneficial to verify whether nLUC and cLUC are translated into proteins in these empty controls or include an unrelated protein with the same subcellular location as suggested in the publication: "Zhou, Z., Bi, G., & Zhou, J.-M. (2018). Luciferase complementation assay for protein-protein interactions in plants. *Current Protocols in Plant Biology*, 3, 42-50. doi: 10.1002/cppb.20066".

Response: Thanks for your suggestions. As you suggested, we used EOG1-nLUC and cLUC-EOG1 as negative controls in this revision (Yan et al., 2024). Consistent with previous conclusion, GS3 can interact with WG1 and itself in *N. benthamiana* leaves (Figure 1C and Figure 2F).

4. The authors claimed that the oligomerization level of GS3 regulates its interaction with RGB1 when treated with DTT, H₂O and H₂O₂ in Figures 3A and 3B. I wonder

how the interaction between GS3 and RGB1 is influenced when WG1 is co-expressed in Yeast Three-Hybrid, LCI or pulldown assays.

Response: Thanks for your suggestions. As you suggested, we investigated the effect of WG1 on the interaction between GS3 and RGB1 using the LCI assay. Co-expression of MYC-WG1 with cLUC-GS3/RGB1-nLUC resulted in a significant enhancement of luciferase activity compared to the negative control (MYC). By contrast, the mutation of the catalytically active sites of WG1 did not lead to a significant change in luciferase activity compared to the control (MYC). These findings indicated that WG1 influences the interaction of GS3 with RGB1 (Figure 3C-D).

5. In Figure 4F, the authors stated that MYC-WG1 reduces the oligomerization level of GFP-GS3 in rice. Is the oligomerization of GS3 increased in the *wg1* mutant (Hao et al., 2021, Molecular Plant)?

Response: Thanks for your suggestions. As you suggested, we transiently transformed the *35S:FLAG-GS3* vector into wild-type (WT) and *wg1-2* mutant protoplasts. Under non-reducing conditions, FLAG-GS3 proteins were found to exist in an oligomeric state in both WT and *wg1-2* mutant protoplasts (Figure EV5). Upon the addition of 4% β -ME (partial reducing), the oligomerization level of FLAG-GS3 was markedly greater in the *wg1-2* mutant than that in the WT background (Figure 4H). This result suggested that the *wg1-2* mutation influences the oligomerization state of GS3.

6. As the authors mentioned, the loss-of-function allele of GS3 has been widely utilized by breeders to improve grain length, but the effect of improving grain length varies among different cultivars. Therefore, I am curious about the natural variation of WG1 among different rice cultivars, landraces and wild rice.

Response: As you suggested, we conducted an analysis of WG1 genomic sequence variation using data from public databases and identified SNPs that are associated with grain length and width in rice (Response Figure 1A, B). Haplotype analysis using these associated SNPs unveiled elite haplotypes linked to grain length and weight, namely Hap5 and Hap4, predominant in *indica* and *japonica* rice, respectively (Response Figure 1C-I), originating from wild rice (Response Figure 1J). Further nucleotide polymorphism analyses and neutrality tests indicated that *WG1* was not under selection during the domestication of wild rice into cultivated rice (Response Table 1). Considering that these results did not significantly contribute to the importance of this study, we did not add these data in this revision.

Minor concerns:

1. In Line 84-87, the authors cited the same publication in different places, but used different indexes. It would be better to maintain consistency in the citation format.

Response: Thanks. We edited this in the revision.

2. WG1 showed weak interaction with GS3 during the seedling stage (Figure 1E-F). According to previous reports, WG1 is expressed at its highest during YP1 stage. Therefore, is it possible to conduct a co-immunoprecipitation (CO-IP) assay for GS3 and WG1 during YP1 stage?

Response: As you suggested, the Co-IP assay was conducted using *pro35S:GFP;pro35S:MYC-WG1* and *pro35S:GFP-GS3;pro35S:MYC-WG1* panicles. The results further supported that GS3 associates with WG1 in panicles (Figure 1E).

3. In Line 128, *N. benthamiana* should be spelled out when used first.

Response: Thanks. We edited this.

4. Line 195, The authors claimed that "Co-expression of Cyfp-GS3 with nYFP-GS3 resulted in strong YFP fluorescence in the plasma membrane and the cytoplasm of epidermal cells in *N. benthamiana* leaves". However, the strong fluorescence dots were observed in Figure 2E. What is the nature of these dots?

Response: To determine the localization of the GS3 self-interaction, we assessed the co-localization of the interaction site with plasma membrane marker (PIP2-mCherry), nuclear marker (H2B-mCherry) and endosome marker (VPS23A-mCherry) in the BiFC assay. As shown in Figure 2G, the signals were observed to partially co-localize with the plasma membrane marker (PIP2-mCherry). The intense fluorescence dots did not co-localize with the nucleus but demonstrated co-localization with the endosome. This result is consistent with a previous study that GS3-2-GFP exhibits strong fluorescence dots in epidermal cells in *N. benthamiana* leaves and co-localizes with endosomes (Yang et al., 2021).

5. In Line 334, WG1 and GS3 should be italic.

Response: We have modified it. Thank you.

Referee #3:

Grain size is crucial for crop yield. The CC-type GRX protein WG1/OsGRX8 has been demonstrated to have disulfide oxidoreductase activity and be involved in

regulation of grain size in rice. However, the specific substrate that WG1 targets to influence grain length remains unclear. In this study, Liu et al., identified GS3, a protein that controls grain length in rice, as an interacting partner of WG1. The authors demonstrated that WG1 physically interacts with GS3 to regulate grain length. They also discovered that GS3 can form oligomers through intermolecular disulfide bonds, which reduce its ability to interact with RGB. Furthermore, WG1 enhances GS3's function in grain length control by decreasing its oligomerization. These findings contribute to our understanding of yield regulation and suggest that WG1 could serve as a molecular target for breeding high-yield rice. However, further experiments are needed to fully validate the authors' conclusions.

1. Both WG1 and GS3 are localized in the nucleus and cytoplasm. However, the authors observed the interaction between WG1 and GS3 only in the plasma membrane and cytoplasm. Furthermore, WG1 is expressed more strongly in the nucleus than in the cytoplasm. Did the authors compare the state of WG1 protein between the nucleus and cytoplasm? This comparison could be significant for understanding its function.

Response: Thanks for your suggestions. As you suggested, we compared the non-reducing state of WG1-GFP proteins in both the nuclear and cytoplasmic fractions. We detected both oxidized WG1-GFP and monomeric WG1-GFP in the nucleus, and oxidized WG1-GFP is predominant, whereas monomeric WG1-GFP is predominantly in the cytoplasm (Appendix Figure S1). This might result in distinct functions of WG1 in the nucleus compared to the cytoplasm.

2. The authors conducted a Co-IP assay to demonstrate the interaction between WG1 and GS3 in transgenic lines, extracting total proteins from 10-day-old seedlings. However, they should provide Co-IP assays from young panicles, as the interaction between WG1 and GS3 is relevant to grain length

Response: As you suggested, the Co-IP assay was performed using *pro35S:GFP;pro35S:MYC-WG1* and *pro35S:GFP-GS3;pro35S:MYC-WG1* panicles. The results supported that GS3 associates with WG1 in panicles (Figure 1E).

3. The authors conducted BIAM labeling assays to demonstrate that GS3 can oligomerize through intermolecular disulfide bonds (Figure 2B). However, this experiment lacks proper controls, which is also a concern in Figures 2F and 4C.

Response: Thanks for your comments. As you suggested, we added proper controls in this revision (Figure 2B, 2H, 4D).

4. The authors indicated that the oligomerization of GS3 weakens its interaction with RGB1. They should provide additional experiments to support this claim.

Response: As you suggested, we further investigated the effect of WG1 on the interaction between GS3 and RGB1 using the LCI assay. Co-expression of MYC-WG1 with cLUC-GS3/RGB1-nLUC resulted in a significant enhancement of luciferase activity compared to the negative control (MYC) (Figure 3C-D). By contrast, the mutations of the catalytically active sites of WG1 did not lead to a significant change in luciferase activity compared to the control (MYC) (Figure 3C-D). These findings further supported that the oligomerization status of GS3 influences its interactions with RGB1.

Yan, L., Jiao, B., Duan, P., Guo, G., Zhang, B., Jiao, W., Zhang, H., Wu, H., Zhang, L., Liang, H., et al. (2024). Control of grain size and weight by the RNA-binding protein EOG1 in rice and wheat. *Cell Rep* 43:114856.

Yang, W., Wu, K., Wang, B., Liu, H., Guo, S., Guo, X., Luo, W., Sun, S., Ouyang, Y., Fu, X., et al. (2021). The RING E3 ligase CLG1 targets GS3 for degradation via the endosome pathway to determine grain size in rice. *Mol Plant* 14:1699-1713.

Dear Prof. Li,

We have now received re-review reports from three referees, which I have included below. As you will see, you have addressed their concerns satisfactorily. Before I can finally accept the manuscript, there are some remaining editorial points which need to be addressed. In this regard would you please:

- remove the AC/CrediT section from the text,
- convert the Appendix file to PDF format; Appendix Figure and Table should be compiled in one Appendix PDF; title page should contain "Appendix for Redox regulation of G protein signaling by WG1 controls grain size in rice" and a table of contents with the page numbers for the listed items; nomenclature should be Appendix Figure Sx and Appendix Table Sx throughout the manuscript and Appendix PDF,
- remove the Reagents and Tools table from the manuscript file, leaving it only as a separate file,
- Figure 1D and Figure 2G contain blank cells in the YFP channel. If the cells are empty or didn't capture any signal then the boxes should have a cross over the appropriate squares,
- provide exact p values in the legends of figures 2E, 3B, 4B and 5D-F,
- define n in the legends of figures 2E, 3B, D; and 4B, E, F, and
- correct the section order as follows: Title page - Abstract & Keywords - Introduction - Results - Discussion - Methods - Data Availability - Acknowledgements - Disclosure and Competing Interests Statement - References - Figure Legends - Table(s) - Expanded View Figure Legends.

We include a synopsis of the paper (see <http://emboj.emboPress.org/>). Please provide me with a general summary image, a two sentence statement and 3-5 bullet points that capture the key findings of the paper. Please also note that our typesetters are able to handle Chinese characters for author names (for an example, see <https://www.emboPress.org/doi/full/10.1038/s44318-024-00147-9>)

I am looking forward to receiving your revised manuscript.

EMBO Press is an editorially independent publishing platform for the development of EMBO scientific publications.

Best wishes,

William Teale

William Teale, PhD
Editor
The EMBO Journal
w.teale@embojournal.org

See also figure legend guidelines: <https://www.emboPress.org/page/journal/14602075/authorguide#figureformat>

- a point-by-point response to the referees' comments, with a detailed description of the changes made (as a word file).
- a word file of the manuscript text.
- individual production quality figure files (one file per figure)
- a complete author checklist, which you can download from our author guidelines (<https://www.emboPress.org/page/journal/14602075/authorguide>).
- Expanded View files (replacing Supplementary Information)

<https://www.emboPress.org/page/journal/14602075/authorguide#expandedview>

- a Reagents and Tools Table as part of the Methods section, which can be downloaded from our author guidelines (<https://www.emboPress.org/page/journal/14602075/authorguide#structuredmethods>)

We realize that it is difficult to revise to a specific deadline. In the interest of protecting the conceptual advance provided by the work, we recommend a revision within 3 months (26th Jun 2025). Please discuss the revision progress ahead of this time with the editor if you require more time to complete the revisions. Use the link below to submit your revision:

Referee #1:

This manuscript provides new and interesting information concerning the mechanism by which the CC-type glutaredoxin WG1/OsGRX8 regulates GS3 oligomerization and grain length in rice. This process is shown to alleviate the inhibitory effect of GS3 on the interaction between RGB1 and DEP1/GGC2, resulting in the increased grain length. The manuscript has been appropriately revised to address all of the points raised.

Referee #2:

The authors have addressed most of my concerns , I have no other comments.

Referee #3:

In this revised manuscript, Liu et al. provided additional evidence to confer their conclusion and address my concerns. Therefore, I would like to support the manuscript to be published.

Dear Editor and Reviewers,

Thank you very much for your supportive and helpful comments on the manuscript (EMBOJ-2024-118898R). We have meticulously addressed all remaining editorial requirements. We added point-by-point response to the editor and reviewers' comments, a general summary image, a two sentence statement and 3 bullet points in this revision. We submitted a formal manuscript text as well as a version with tracked changes.

Editorial points which need to be addressed:

- remove the AC/CrediT section from the text,

Response: We removed the AC/CrediT section from the text in the revision.

- convert the Appendix file to PDF format; Appendix Figure and Table should be compiled in one Appendix PDF; title page should contain "Appendix for Redox regulation of G protein signaling by WG1 controls grain size in rice" and a table of contents with the page numbers for the listed items; nomenclature should be Appendix Figure Sx and Appendix Table Sx throughout the manuscript and Appendix PDF,

Response: As you suggested, all appendix figures and tables have been consolidated into a single PDF file, accompanied by a dedicated title page and table of contents. The nomenclature has been standardized throughout (Appendix Figure S1, Appendix Table S1) to ensure full compliance with the requested format.

- remove the Reagents and Tools table from the manuscript file, leaving it only as a separate file,

Response: The Reagents and Tools Table has been removed from the manuscript and provided as a separate file.

- Figure 1D and Figure 2G contain blank cells in the YFP channel. If the cells are empty or didn't capture any signal then the boxes should have a cross over the appropriate squares,

Response: Many thanks for your kind reminder. In Figures 1D and 2G, we showed multi-channel confocal images (YFP, mCherry, and bright-field merged views), which were carefully selected to encompass representative fields of view. These images unambiguously demonstrate cell-specific YFP signal presence or absence in the indicated combinations.

- provide exact p values in the legends of figures 2E, 3B, 4B and 5D-F,

Response: In this revision, we showed p-values of Figures 2E, 3B, 4B and 5D-F in the figure legends. The p-value was generated using GraphPad Prism software with appropriate statistical analysis. When the p-value is below 0.0001 or above 0.9999, this software displays '<0.0001' or '>0.9999' instead of the exact value.

- define n in the legends of figures 2E, 3B, D; and 4B, E, F, and

Response: We added n in the legends of Figures 2E, 3B, D and 4B, E, F.

- correct the section order as follows: Title page - Abstract & Keywords - Introduction - Results - Discussion - Methods - Data Availability - Acknowledgements - Disclosure and Competing Interests Statement - References - Figure Legends - Table(s) - Expanded View Figure Legends.

Response: Thanks. We have reorganized the sections in order.

Referee #1:

This manuscript provides new and interesting information concerning the mechanism by which the CC-type glutaredoxin WG1/OsGRX8 regulates GS3 oligomerization and grain length in rice. This process is shown to alleviate the inhibitory effect of GS3 on the interaction between RGB1 and DEP1/GGC2, resulting in the increased grain length. The manuscript has been appropriately revised to address all of the points raised.

Response: Thanks.

Referee #2:

The authors have addressed most of my concerns, I have no other comments.

Response: Thanks.

Referee #3:

In this revised manuscript, Liu et al. provided additional evidence to confer their conclusion and address my concerns. Therefore, I would like to support the manuscript to be published.

Response: Thanks.

Yours sincerely,

Yunhai Li, Corresponding Author

yhli@genetics.ac.cn

Tel: 86-10-64807856

Dear Yunhai,

I am pleased to inform you that your manuscript has been accepted for publication in the EMBO Journal.

Congratulations!

Best wishes,

William

William Teale, PhD
Editor
The EMBO Journal
w.teale@embojournal.org
